# Maladaptive cardiomyocyte calcium handling in adult offspring of hypoxic pregnancy: protection by antenatal maternal melatonin

Mitchell C. Lock[1] , Olga V. Patey[2] , Kerri L. M. Smith[1] , Youguo Niu[2] , Ben Jaggs[2], Andrew W. Trafford[1] , Dino A. Giussani[2] and Gina L. J. Galli[1] 

[1] *Division of Cardiovascular Sciences, School of Medical Sciences, University of Manchester, Manchester, UK*
[2] *Department of Physiology, Development and Neuroscience, University of Cambridge, Cambridge, UK*

Handling Editors: Laura Bennet & Christopher Lear

The peer review history is available in the Supporting Information section of this article (https://doi.org/10.1113/JP287325#support-information-section).

**Abstract figure legend** Wistar rats were time-mated and exposed to either normoxia or hypoxia (13% oxygen), with or without melatonin treatment from gestational day (GD) 6 to 20. The cardiac impact was investigated in offspring at 4 months of age. Hypoxia caused increased sphericity index and diastolic $Ca^{2+}$ dysfunction, altered ventricular wall thickness as well as decreased systolic and diastolic function, which was prevented by melatonin treatment.

 

**Mitchell C. Lock** is an early career researcher within the Early Origins of Adult Health Research Group at the University of South Australia. His research focuses on understanding the molecular mechanisms by which sub-optimal *in utero* conditions such as fetal hypoxaemia can influence lung maturation and cardiac development and the programming of adult cardiovascular disease. More recently he has been focused on investigating the therapeutic potential of antenatal antioxidants and how their use in hypoxic pregnancy may provide protection for the lung and heart in both early postnatal life and adulthood. **Olga Patey** is a fetal cardiologist at the Royal Brompton Hospital and St. George's University Hospital in London, UK. Dr Patey is deeply committed to advancing experimental research and is actively engaged in exploring fetal, maternal, and offspring cardiovascular health using various animal models at the University of Cambridge, UK. Additionally, she serves as a dedicated clinical researcher at the Nuffield Department of Women's & Reproductive Health, University of Oxford, UK, where she focuses on the application of artificial intelligence in fetal echocardiography to enable early detection of congenital heart defects in low- and middle-income countries. Her fluency in English, Russian, and Arabic allows her to effectively communicate and collaborate on a global scale.

M. C. Lock, O. V. Patey, D. A Giussani and G. L. J. Galli contributed equally to this work.

The Journal of Physiology

**Abstract** Chronic fetal hypoxia is one of the most common complications of pregnancy and can programme cardiac abnormalities in adult offspring including ventricular remodelling, diastolic dysfunction and sympathetic dominance. However, the underlying mechanisms at the level of the cardiomyocyte are unknown, preventing the identification of targets for therapeutic intervention. Therefore, we aimed to link echocardiographic data with cardiomyocyte function to reveal cellular mechanism for cardiac dysfunction in rat offspring from hypoxic pregnancy. Further, we investigated the potential of maternal treatment with melatonin as antenatal antioxidant therapy. Wistar rats were randomly allocated into normoxic (21% $O_2$) or hypoxic (13% $O_2$) pregnancy with or without melatonin treatment (5 µg/ml; normoxic melatonin in the maternal drinking water from gestational day 6 to 20 (term = 22 days). After delivery, male and female offspring were maintained to adulthood (16 weeks). Cardiomyocytes were isolated from the left and right ventricles, and calcium ($Ca^{2+}$) handling was investigated in field-stimulated myocytes. Systolic and diastolic function was negatively impacted in male and female offspring of hypoxic pregnancy demonstrating biventricular systolic and diastolic dysfunction and compensatory increases in cardiac output. $Ca^{2+}$ transients from isolated cardiomyocytes in offspring of both sexes in hypoxic pregnancy displayed diastolic dysfunction with a reduced rate of $[Ca^{2+}]_i$ recovery. Cardiac and cardiomyocyte dysfunction in male and female adult offspring was ameliorated by maternal antenatal treatment with melatonin in hypoxic pregnancy. Therefore, cardiomyocyte $Ca^{2+}$ mishandling provides a cellular mechanism explaining functional deficits in hearts of male and female offspring in pregnancies complicated by chronic fetal hypoxia.

(Received 17 July 2024; accepted after revision 10 October 2024; first published online 21 November 2024)
**Corresponding author** M. C. Lock: Early Origins of Adult Health Research Group, HIB Level 5, UniSA, Clinical and Health Sciences, University of South Australia.    Email: Mitchell.Lock@unisa.edu.au

## Key points

- This study identified significant changes in $Ca^{2+}$ handling within cardiomyocytes isolated from offspring of hypoxic pregnancy including reduced systolic $Ca^{2+}$ transients, impaired diastolic recovery of $[Ca^{2+}]_i$ and a greater increase in systolic $[Ca^{2+}]_i$ amplitude to $\beta$-adrenergic stimulation.
- These changes in cardiomyocyte $Ca^{2+}$ handling help to explain dysregulation of biventricular systolic and diastolic dysfunction determined by echocardiography.
- The data show protection against maladaptive cardiomyocyte calcium handling and thereby improvement in cardiac function in adult offspring of hypoxic pregnancy treated with melatonin with doses lower than those recommended for overcoming jet lag in humans.
- Melatonin treatment alone in healthy pregnancy did cause some alterations in cardiac structure. Therefore, maternal treatment with melatonin should only be given to pregnancies affected by chronic fetal hypoxia.

## Introduction

Cardiovascular disease is the leading cause of morbidity and mortality worldwide with an estimated 17.9 million deaths in 2019 (WHO, 2021). It is well-established that genetics and lifestyle factors play a major role in the development of cardiovascular disease. However, cardiovascular disease can also be driven by exposure to environmental stress, especially when it occurs during fetal development. This can be mediated by epigenetic changes to gene expression persisting through the life-time. Indeed, human and animal studies have shown that sub-optimal intrauterine conditions can lead to fetal cardiac abnormalities and increased rates of heart disease in adulthood (Gluckman et al., 2008). Therefore, a window of opportunity exists for preventing programmed cardiovascular risk as early as possible, for instance during fetal development (Giussani, 2021).

A sustained reduction in fetal oxygenation or chronic fetal hypoxia is one of the most common outcomes in complicated pregnancy (Giussani, 2016). This condition can result from numerous maternal, placental or fetal

complications, including pregnancy at high altitude, maternal smoking, preeclampsia, placental insufficiency, or umbilical cord compression (Hutter et al., 2010). The fetus initially responds to hypoxia by preferentially distributing blood flow to vital organs, such as the brain – the fetal brain-sparing effect (Cohn et al., 1974; Giussani, 2016; Giussani et al., 1993). Although protective in the short-term, preclinical animal models have shown that this strategy can lead to increased fetal peripheral vascular resistance, raising fetal cardiac afterload and triggering fetal ventricular hypertrophy (Ding et al., 2020; Giussani et al., 1993; Hauton & Ousley, 2009; Spiroski et al., 2021). Importantly, in rats, offspring from hypoxic pregnancies continue to display cardiac abnormalities in adulthood, including left ventricular (LV) wall thickening (Rueda-Clausen et al., 2008) and LV diastolic dysfunction (Aljunaidy et al., 2018; Niu et al., 2018; Rueda-Clausen et al., 2008; Spiroski et al., 2021). *In vivo* LV systolic function appears to be preserved in offspring from hypoxic rat pregnancies (Rueda-Clausen et al., 2008), resulting from an increase in sympathetic activity and LV contractility, maintaining cardiac output (Hauton & Ousley, 2009; Niu et al., 2018). Comparative measurements in the right ventricle (RV) are lacking, but several studies in rats, chicken and sheep suggest offspring from hypoxic pregnancies develop signs of pulmonary hypertension, including pulmonary arterial wall thickening, decreased pulmonary artery acceleration time and increased RV diameter in diastole (Botting et al., 2020; Ding et al., 2020; Li et al., 2021; Rueda-Clausen et al., 2008; Skeffington et al., 2020). Chronic fetal hypoxia also sensitizes the rat heart to ischaemia–reperfusion injury at 3 months of age (Xue & Zhang, 2009), which may predispose to myocardial infarction. The best evidence of cardiovascular dysfunction in offspring from hypoxic pregnancies in humans comes from pregnancies complicated by chronic fetal hypoxia leading to fetal growth restriction (FGR). Reported abnormalities in cardiac morphology and function in FGR human fetuses include ventricular wall hypertrophy at 17–24 weeks (Veille et al., 1993), compromised ejection force at 18–38 weeks (Rizzo et al., 1995) and impaired diastolic filling (Miyague et al., 1997). In addition, human clinical studies have reported that low birth weight is associated with globular cardiac ventricles with impaired stroke volume in children ($\sim$5 years old), and endothelial dysfunction and systemic hypertension by young adulthood (20–28 years old) (Bjarnegård et al., 2013; Crispi et al., 2010; Leeson et al., 2001). Taken together, these data suggest that adult offspring from hypoxic pregnancies have early signs of biventricular dysfunction, increasing cardiovascular risk. Interestingly, the effects of chronic fetal hypoxia are strongly sex-dependent, with males being more susceptible to cardiac developmental programming than females (Intapad et al., 2014; Padhee et al., 2023; Rueda-Clausen et al., 2008).

While the link between chronic fetal hypoxia and offspring cardiac abnormalities in adulthood is well-established, the cellular mechanisms driving the phenotype are unclear. Diastolic and systolic dysfunction could be underpinned by abnormal cardiomyocyte calcium cycling. The magnitude of the systolic rise in $[Ca^{2+}]_i$ is the primary determinant of contractile force in the ventricle, meaning the increase in LV contractility in offspring from hypoxic pregnancies could be driven by greater cardiomyocyte $Ca^{2+}$ influx during contraction. Further, diastolic dysfunction often occurs at the cellular level due to impaired $Ca^{2+}$ removal, which prolongs the $[Ca^{2+}]_i$ decay. In addition, the enhanced cardiac sympathetic dominance in offspring from hypoxic pregnancies may also have a cellular origin. $\beta$-Adrenergic receptor stimulation is a major pathway for raising systolic $[Ca^{2+}]_i$ (Bers, 2001). Studies in chicken embryos have reported that chronic hypoxia increases cardiac $\beta$-adrenergic receptor density (Lindgren & Altimiras, 2013), but investigation of the interaction between sympathetic signalling and changes in $[Ca^{2+}]_i$ in any species is also lacking.

In recent years, maternal antioxidant treatments have gained significant interest in preventing oxidative stress in at-risk pregnancies. Interventional studies in preclinical animal models have demonstrated prevention of cardiovascular dysfunction by allopurinol, vitamin C, MitoQ and melatonin (Botting et al., 2020; Giussani et al., 2012; Hansell et al., 2022; Niu et al., 2018). Melatonin, in particular, has become a prime candidate to protect the fetus from oxidative stress because of its low toxicity and high efficiency, as well as its protective roles in cardiac and neural development (Lemley et al., 2012; Richter et al., 2009). Importantly, we have recently demonstrated the effectiveness of maternal treatment with melatonin in preventing cardiac oxidative stress in rodent hypoxic pregnancy and there are ongoing human clinical trials for use of melatonin for antioxidant protection in high-risk pregnancies (Hansell et al., 2022; Hobson et al., 2018). However, whether maternal melatonin treatment can protect against the developmental programming of excitation–contraction pathway dysfunction within the cardiomyocyte under basal or stimulated conditions in offspring of hypoxic pregnancy remains completely unknown.

In this study, we tested the hypothesis that the mechanism of programmed cardiac dysfunction in offspring from hypoxic pregnancies is underpinned by altered cardiomyocyte $Ca^{2+}$ homeostasis. The hypothesis was tested using an established rat model of hypoxic pregnancy linking echocardiographic data with measurements of intracellular $Ca^{2+}$ transients in

ventricular myocytes isolated from adult offspring. We also investigated the effectiveness of maternal melatonin treatment as an antenatal antioxidant on cardiac and cardiomyocyte dysfunction. Given strong sex-dependent effects of developmental programming (Sandovici et al., 2022), we performed experiments on male and female offspring.

## Methods

### Animal ethics

This study was approved by the University of Manchester and Cambridge University Animal Welfare and Ethical Review Board (PD7C22AA9). All procedures were carried out under the UK Animals (Scientific Procedures) Act 1986. All investigators understood and followed the ethical principles outlined by Grundy (2015) and the principles of the 3Rs, specifically the reduction of the use of animals in research (Russell & Burch, 1959). Experiments were designed and reported with reference to the ARRIVE guidelines (Kilkenny et al., 2010) and conform to the guidelines from Directive 2010/63/EU of the European Parliament on the protection of animals for scientific purposes and the NIH *Guide for the Care and Use of Laboratory Animals.*

### Animals and experimental design

Wistar rats (Charles River Laboratories, Saffron Walden, UK) were delivered to the University of Manchester Biological Services Facility and housed in individually ventilated cages (IVC units, 21% $O_2$, 70–80 air changes per hour) in rooms with controlled humidity (60%), controlled temperature (21°C) and a 12:12 h light–dark cycle with free access to food and water (Envigo, Indianapolis, IN, USA). After 1 week of acclimatization, virgin female Wistar rats weighing between 225 and 250 g were paired individually with male Wistar rats. Cages were checked daily, and the appearance of the copulatory plug was taken to be day 0 of pregnancy (term ∼22 days). On gestational day (GD) 6, pregnant rats were randomly allocated to one of four treatment groups: Normoxia (N), Normoxia + Melatonin (NM), Hypoxia (H) and Hypoxia + Melatonin (HM). Maternal weight and food and water consumption were monitored every 48 h.

Pregnant rats assigned to hypoxia groups were placed inside a transparent hypoxic chamber (Coy Laboratory Products, Inc., Grass Lake, MI, USA), as previously described (Hellgren et al., 2021). The chamber consists of a clear PVC isolator attached to nitrogen and oxygen cylinders to precisely control the percentage of oxygen within the chamber. Carbon dioxide and atmospheric waste were scavenged by circulating the air through soda lime pellets and activated charcoal pellets, respectively (Sigma-Aldrich, St Louis, MO, USA). The chamber contained a hygrometer and thermometer for continuous monitoring of humidity and temperature whilst oxygen was monitored with the inbuilt calibrated oxygen sensor (Coy Laboratory Products, Inc.). Pregnancies undergoing maternal hypoxia were maintained at an inspired fraction of oxygen of 13% from GD6 to GD20 to model early-onset chronic fetal hypoxia, and food and water consumption were recorded every 48 h. This level of hypoxia is human clinically relevant as it is equivalent to *ca.* 3650 m of high altitude, when human pregnancy complications significantly increase (Grant et al., 2021; Salinas et al., 2023; Soria et al., 2013). This level of maternal hypoxia also leads to reductions in fetal $P_{O_2}$ measured by cordocentesis in human infants from pregnancy complicated by fetal growth restriction (Allison et al., 2020; Nicolaides et al., 1986). The onset of hypoxia in rat pregnancy was on GD6, as studies have revealed markedly enhanced pregnancy loss if the hypoxia started earlier (Giussani et al., 2012).

Animals assigned to the melatonin group received melatonin (Sigma-Aldrich) within their drinking water at a concentration of 5 µg/ml dissolved in minimum required ethanol (150 µl in 500 ml water). The final concentration of ethanol in the water was <0.0003%. The control group animals received the equivalent amount of ethanol without melatonin in their water. Drinking water was prepared fresh every 48 h. The dose of melatonin used was equivalent to the maximal dose recommended for overcoming jet lag in humans (Herxheimer & Petrie, 2002), and has previously been used by our group (Hansell et al., 2022; Smith et al., 2022). Changes in plasma melatonin concentrations have previously been published in this model (Smith et al., 2022) and reached therapeutic levels in both maternal and fetal plasma samples.

At GD20, rats were returned to normal IVC cages for birth at GD22. At birth each litter was reduced to 8 pups (4 males and 4 females, randomized with the exception of pups that were unlikely to survive to weaning) to standardize nutritional intake. At weaning males and females were separated and housed as groups of four siblings until 16–19 weeks of age. At this time, one male and one female offspring from each litter were randomly assigned to undergo either cardiomyocyte isolation ($n = 50$) or echocardiography ($n = 69$) experiments.

### Cardiomyocyte isolation in adult offspring

Animals assigned to the cardiomyocyte isolation group were humanely killed with $CO_2$ inhalation followed by cervical spinal transection. The heart was then quickly removed and mounted onto a Langendorff apparatus

retrogradely perfusing through the aorta with a nominally $Ca^{2+}$-free solution (mmol/l: NaCl 134, glucose 11.1, HEPES 10, KCl 4, $MgSO_4$ 1.2 and $NaH_2PO_4$ 1.2, pH 7.34 with NaOH) for 5 min at 37°C. Liberase (Roche, Switzerland) at 6 Wünsch units was added to 50 ml $Ca^{2+}$-free solution and perfused for 10–13 min until the heart was visibly flaccid. The perfusate was then changed to a taurine containing solution (mmol/l: NaCl 115, glucose 11.1, HEPES 10, KCl 4, $MgSO_4$ 1.2, $NaH_2PO_4$ 1.2 and taurine 50, pH 7.34 with NaOH) for 10 min. The heart was then dissected; left ventricle and right ventricle were separated and cut into small pieces in a taurine solution and gently triturated to liberate single cells. The isolated cells were stored at room temperature in a taurine-containing solution until use.

## Measurement of cardiomyocyte intracellular $Ca^{2+}$ ($[Ca^{2+}]_i$)

Cells were loaded with the $Ca^{2+}$ sensitive fluorescent indicator Fura-2 AM in dimethyl sulfoxide/Pluronic stock at 1 µmol/l in for 5 min in a Tyrode solution containing (mmol/l) NaCl 140, HEPES 10, glucose 10, Probenecid 2, KCl 4, $MgCl_2$ 1 and $CaCl_2$ 1, pH 7.34 with NaOH. Loaded cells were then pipetted onto a Nikon (Tokyo, Japan) Eclipse TE2000-U microscope with a Nikon Plan Flour ×40 oil immersion objective lens fitted with a MPRE8 inline heated tip (Cell MicroControls, Norfolk, VA, USA) with constant perfusion of Tyrode solution at 37°C. Individual cardiomyocytes were located and sequentially excited at 340 nm and 380 nm. Cardiomyocytes were field-stimulated at 1 Hz and $[Ca^{2+}]_i$ was recorded at 1 kHz sampling rate for 5 min until $Ca^{2+}$ transients were stabilized. The sarcoplasmic reticulum $Ca^{2+}$ content was quantified by exposing cells to 10 mM caffeine (Sigma-Aldrich), as previously described (Clarke et al., 2015). In addition, the cellular response to $\beta$-adrenergic stimulation was induced by exposure to 100 nmol $l^{-1}$ isoprenaline sulfate (Stockport Pharmaceuticals, Stepping Hill Hospital, Stockport, UK). The settings remained the same throughout all experimental groups and background fluorescence was subtracted from all signals.

## Cardiomyocyte $Ca^{2+}$ transient analysis

One investigator blinded to the treatment grounds analysed $Ca^{2+}$ transients using Clampfit 11 software (Molecular Devices, San Jose, CA, USA). For each $Ca^{2+}$ transient the $F_{340/380}$ ratio was calculated for: diastolic, systolic, amplitude, decay time (DT)50, DT90, and the decay time constant (tau) of the $Ca^{2+}$ transient. Each measurement represents the average of five successive transients from each cell electrically paced at steady state.

## Measurement of cardiomyocyte contractility

Video recordings of stimulated cells were made using a CCD camera (W82C CCIR, Watec Cameras, Pine Bush, NY, USA) at 30 frames/s using GrabBee 2.0 video recording software. Video recordings were then shortened to 1-min clips of cells at steady state before and after isoprenaline treatment in uncompressed AVI format using Adobe Premiere Pro 2020 (Abobe Systems, San Jose, CA, USA). Videos were then imported into ImageJ software (version 1.52s), and analysed using the Myocyter v1.3 macro (Grune et al., 2019). Using Myocyter, we calculated the relative amplitude of contraction, representing the normalized extent of cell shortening during contraction (mean amplitude), contraction/relaxation times (s) for time to peak, time to 50% systole, time to 50% diastole, and mean velocity of contraction/relaxation (mean amplitude/systolic and diastolic times).

## Echocardiography

In a separate cohort of adult offspring, two-dimensional (2D), spectral or pulsed-wave (PW) Doppler and PW-tissue Doppler imaging (TDI) images were acquired using a Vivid-iQ (GE Healthcare, Milwaukee, WI, USA) ultrasound system. Isoflurane was used to induce sedation and immobility. In brief, the animal was placed in an oxygenated box and then the vaporizer was turned to 5% isoflurane, which rapidly filled the box with a rising concentration of anaesthetic. Once the animal was under a deep and stable anaesthesia, it was moved to a mask with the isoflurane on either 2.5% or 3% while the animal was shaved and cleansed. Echocardiography was performed with the animal under 2.5% isoflurane. The amount of anaesthetic needed was minimized during the echocardiography procedure. Following echocardiography, the animals were placed into a heated cabinet for recovery and regaining of consciousness.

One investigator blinded to the treatment groups performed the analysis on vendor-specific software EchoPAC (Viewpoint, version 112). The following techniques and cardiac parameters were utilized: (1) 2D measurements of ventricular dimensions and areas, valve size, and calculation of ventricular sphericity indices (SI) and fractional area change (FAC) (Patey et al., 2019); (2) PW Doppler from the inflow and outflow tracts to evaluate diastolic and systolic function, respectively, and calculate left ventricular (LV) and RV stroke volume (SV), cardiac output (CO), and combined CO (CCO); additionally, PW Doppler was used to measure peak systolic velocity and pulsatility index in the carotid artery, ascending aorta, descending aorta and femoral artery; (3) PW-TDI to investigate cardiac indices of myocardial motion in systole and diastole, myocardial time intervals,

and for estimation of LV and RV myocardial performance index (MPI′); and (4) speckle tracking echocardiography (STE) to derive ventricular indices of longitudinal strain and systolic and diastolic strain rate and other aspects of ventricular systolic and diastolic myocardial deformation (circumferential, radial and rotational). The global strain was reported as an absolute value. The time-interval values were calculated as a proportion of cardiac cycle length normalising for differences in heart rate (Patey et al., 2019). All echocardiographic measurements were performed according to the standardized protocol of the study and with regards to previously described echo techniques (Patey et al., 2019).

### Statistical analysis

Statistical analyses were performed using SAS (SAS Institute Inc., Cary, NC, USA) and SPSS Statistics (IBM Corp., Armonk, NY, USA). Data in SAS were separated by ventricle and analysed using a mixed linear model. The model included nested measurements within an animal, by calculating two components of variability: the between animal and the within animal data. The main effects of oxygen/treatment/sex were assessed simultaneously, followed with oxygen × treatment, treatment × sex, and oxygen × sex first order interactions, and with the second order interaction of oxygen × treatment × sex. Data in SPSS were analysed using a nested two-way ANOVA for multiple cells isolated and analysed from each animal. For all comparisons, statistical significance was accepted when $P < 0.05$.

### Results

#### Sex-dependent effects

The majority of the results did not reveal a sex-dependent effect; thus, a majority of outcomes were combined and the sex of the rats is denoted in the graphs by different symbols (Figs 1, 3–6). Mixed model analysis detected only three sex-dependent effects within our entire dataset. The three sex-dependent effects are as follows. First, there was a lower $Ca^{2+}$ transient amplitude from RV-isolated cardiomyocytes of female but not male offspring of hypoxic pregnancy ($P = 0.025$). Secondly, there was an increased DT50 in cardiomyocytes isolated from female offspring of melatonin-treated pregnancies after isoprenaline treatment ($P = 0.031$). Finally, RV relative wall thickness (RWT) was increased by a greater margin in male compared with female offspring of hypoxic pregnancy, but this measure was only significant when compared to normoxic offspring when sexes were combined (Table 1).

### Cardiac function measured by echocardiography

**Effects of chronic fetal hypoxia.** Systolic and diastolic cardiac function was significantly altered in offspring from hypoxic pregnancies, compared to controls. Cardiac output (CO) and stroke volume were increased (Fig. 1*A* and Table 1; interaction, $P < 0.001$), basal radial late diastolic strain rate was reduced (Fig. 1*B*; main effect, $P = 0.0023$) and deceleration time was increased (Table 1; main effect, $P < 0.001$) in the LV of adult offspring of hypoxic pregnancy. Diastolic dysfunction was also present in the RV of adult offspring from hypoxic pregnancies with an increase in early ($E′$) and late ($A′$) diastolic myocardial velocities and a decrease in diastolic $E/E′$ ratio (Table 1; main effect, $P = 0.028$). Within both the LV and RV of adult offspring myocardial performance index (MPI′), isovolumetric contraction time (IVCT′) and isovolumetric relaxation time (IVRT′) were increased in offspring exposed to chronic fetal hypoxia (Fig. 1*C–H*; interaction, $P = 0.025$, $P = 0.016$, $P = 0.009$, $P = 0.005$, $P = 0.03$ and $P = 0.001$, respectively), but had no effect on the RV fractional area change (FAC). Maximum velocity ($V_{max}$) in the ascending aorta (AAo) was also increased in adult offspring of hypoxic pregnancy (Fig. 1*I*; main effect, $P < 0.042$). Pulsatility index (PI) was unaltered in the ascending aorta (AAo), but was increased in the carotid artery (CCA) in adult offspring of hypoxic pregnancy (Table 1; main effect, $P = 0.002$).

**Effects of maternal melatonin treatment.** Maternal antenatal melatonin treatment prevented all of the measured functional changes in the LV and RV in adult offspring from hypoxic pregnancies, except for the LV radial late diastolic strain rate, RV IVRT′ (Fig. 1*B* and *H*; main effect, $P = 0.0023$, $P < 0.001$), and the AAo $V_{max}$ (Fig. 1*I*; main effect, $P = 0.042$), which remained unchanged. Maternal melatonin in the normoxic group revealed a reduced LV diastolic $E/A$ ratio, LV radial late diastolic strain rate, and increased AAo PI in adult offspring (Table 1; main effect, $P = 0.0438$; Fig. 1*B*; main effect, $P = 0.048$; Table 1; main effect, $P = 0.013$).

### Cardiac structure measured by echocardiography

**Effects of chronic fetal hypoxia.** There was evidence of cardiac remodelling in the LV and RV in offspring from hypoxic pregnancies. Within the LV there was reduced relative wall thickness (Table 1; main effect, $P = 0.047$), increased sphericity index (Table 1; main effect, $P = 0.002$), but no effect on end diastolic diameter (LVEDD) (Table 1) in animals exposed to chronic fetal hypoxia. Within the RV relative wall thickness was increased in adult offspring of chronic fetal hypoxia (Table 1; main effect, $P < 0.001$). The aortic valve to pulmonary valve dimension ratio (AV/PV ratio) in adult

**Table 1. Cardiac structure and function measured by echocardiography**

| Echocardiographic parameters | Normoxia (n = 13) | Normoxia melatonin (n = 20) | Hypoxia (n = 18) | Hypoxia melatonin (n = 18) |
|---|---|---|---|---|
| **Cardiac structure** | | | | |
| LV end-diastolic dimension (mm) | 7.23 ± 0.73 | 7.05 ± 0.89 | 7.81 ± 1.18 | 7.25 ± 1.18 |
| RV end-diastolic dimension (mm) | 3.69 ± 0.63 | 3.55 ± 0.51 | 3.83 ± 0.62 | 3.57 ± 0.85 |
| LV sphericity index | 0.66 ± 0.07 | 0.64 ± 0.09 | 0.74 ± 0.10** | 0.71 ± 0.07** |
| RV sphericity index | 0.37 ± 0.06 | 0.39 ± 0.07 | 0.40 ± 0.07 | 0.36 ± 0.09 |
| LV relative wall thickness | 0.47 ± 0.14 | 0.57 ± 0.07 | 0.43 ± 0.13* | 0.47 ± 0.10 |
| RV relative wall thickness | 0.63 ± 0.19 | 0.58 ± 0.09 | 0.82 ± 0.25** | 0.82 ± 0.32** |
| Aortic valve/pulmonary valve dimension ratio | 0.96 ± 0.15 | 0.92 ± 0.21 | 1.25 ± 0.28* | 1.00 ±0.13 |
| Mitral valve/tricuspid valve dimension ratio | 1.42 ± 0.48 | 1.27 ± 0.28† | 1.54 ± 0.41 | 1.33 ± 0.27† |
| **Vasculature** | | | | |
| Ascending aorta pulsatility index | 2.69 ± 0.41 | 2.91 ± 0.22† | 2.77 ± 0.32 | 2.95 ± 0.35† |
| Carotid artery pulsatility index | 3.12 ± 0.44 | 2.86 ± 0.20 | 3.24 ± 0.39* | 3.26 ± 0.36* |
| Carotid artery maximal velocity $V_{max}$ (cm/s) | 82.26 ± 19.58 | 94.77 ± 22.33 | 89.28 ± 14.37 | 96.31 ± 19.69 |
| **Systolic function** | | | | |
| Heart rate (bpm) | 373 ± 43 | 384 ± 31 | 365 ± 39* | 360 ± 44 |
| Cardiac output (ml/min/kg) | 321 ± 80 | 315 ± 49 | 468v98* | 354 ± 49† |
| LV fractional shortening (%) | 40.29 ± 4.67 | 43.07 ± 7.22 | 41.65 ± 8.16 | 39.97 ± 6.87 |
| LV ejection fraction (%) | 81.13 ± 4.29 | 77.70 ± 7.01 | 78.31 ± 9.31 | 72.33 ± 6.85 |
| LV stroke volume (ml/min/kg) | 0.92 ± 0.25 | 0.98 ± 0.23 | 1.47 ± 0.28** | 1.04 ± 0.23 |
| LV myocardial systolic velocity $S'$ (cm/s) | 5.56 ± 2.06 | 4.95 ± 1.09 | 6.08 ± 1.69 | 5.11 ± 2.18 |
| RV myocardial systolic velocity $S'$ (cm/s) | 5.27 ± 0.61 | 6.37 ± 1.76 | 6.78 ± 1.83 | 6.20 ± 1.15 |
| LV ejection time ET$'$ (ms) | 0.40 ± 0.07 | 0.41 ± 0.04 | 0.41 ± 0.05 | 0.43 ± 0.06 |
| RV ejection time ET$'$ (ms) | 0.43 ± 0.03 | 0.41 ± 0.06 | 0.40 ± 0.04 | 0.44 ± 0.03 |
| **Diastolic function** | | | | |
| LV deceleration time (ms) | 36.92 ± 10.81 | 39.20 ± 10.87 | 48.53 ± 13.11** | 50.72± 14.86** |
| LV early diastolic transvalvar velocity E (cm/s) | 94.83 ± 28.51 | 99.13 ± 22.44 | 98.97 ± 12.92 | 100.94 ± 13.41 |
| LV late diastolic transvalvar velocity A (cm/s) | 89.17 ± 37.51 | 100.58 ± 18.32 | 97.29 ± 17.55 | 99.39 ± 15.80 |
| LV diastolic E/A ratio | 1.21 ± 0.51 | 0.98 ± 0.11† | 1.04 ± 0.17 | 1.02 ± 0.10† |
| LV early diastolic myocardial velocity $E'$ (cm/s) | 8.03 ± 3.61 | 8.95 ± 4.66 | 6.92 ± 1.97 | 7.33 ± 3.05 |
| LV late diastolic myocardial velocity $A'$ (cm/s) | 8.41 ± 3.34 | 9.22 ± 4.47 | 7.16 ± 1.43 | 7.84 ± 2.96 |
| LV diastolic $E'/A'$ ratio | 0.96 ± 0.20 | 0.96 ± 0.11 | 0.96 ± 0.15 | 0.94 ± 0.16 |
| LV diastolic $E/E'$ ratio | 13.79 ± 10.49 | 12.86 ± 5.19 | 15.80 ± 5.04 | 15.55 ± 4.23 |
| RV early diastolic transvalvar velocity E (cm/s) | 69.75 ± 22.33 | 75.44 ± 12.41 | 70.44 ± 20.35 | 69.33 ± 13.16 |
| RV late diastolic transvalvar velocity A (cm/s) | 69.75 ± 22.33 | 70.44 ± 19.14 | 73.47 ± 18.97 | 71.28 ± 11.53 |
| RV diastolic E/A ratio | 1.00 ± 0.00 | 1.16 ± 0.44 | 0.98 ± 0.25 | 0.98 ± 0.11 |
| RV early diastolic myocardial velocity $E'$ (cm/s) | 7.06 ± 2.66 | 9.78 ± 3.61 | 10.59 ± 3.00* | 10.27 ± 3.30* |
| RV late diastolic myocardial velocity $A'$ (cm/s) | 7.15 ± 2.64 | 9.78 ± 3.61 | 11.02 ± 3.13** | 10.44 ± 3.11** |
| RV diastolic $E'/A'$ ratio | 0.99 ± 0.04 | 1.00 ± 0.00 | 0.97 ± 0.15 | 0.99 ± 0.14 |
| RV diastolic $E/E'$ ratio | 10.96 ± 4.36 | 8.34 ± 4.92 | 6.61 ± 2.06* | 7.84 ± 4.19* |

Values are means ± SD. Data were analysed using a mixed linear model. The model included nested measurements within an animal, $n = 69$ animals total. The main effects of oxygen/treatment/sex were assessed simultaneously. Hypoxia effect: *$P < 0.05$, **$P < 0.01$. Melatonin effect: †$P < 0.05$, ††$P < 0.01$. LV, left ventricular; RV, right ventricular.

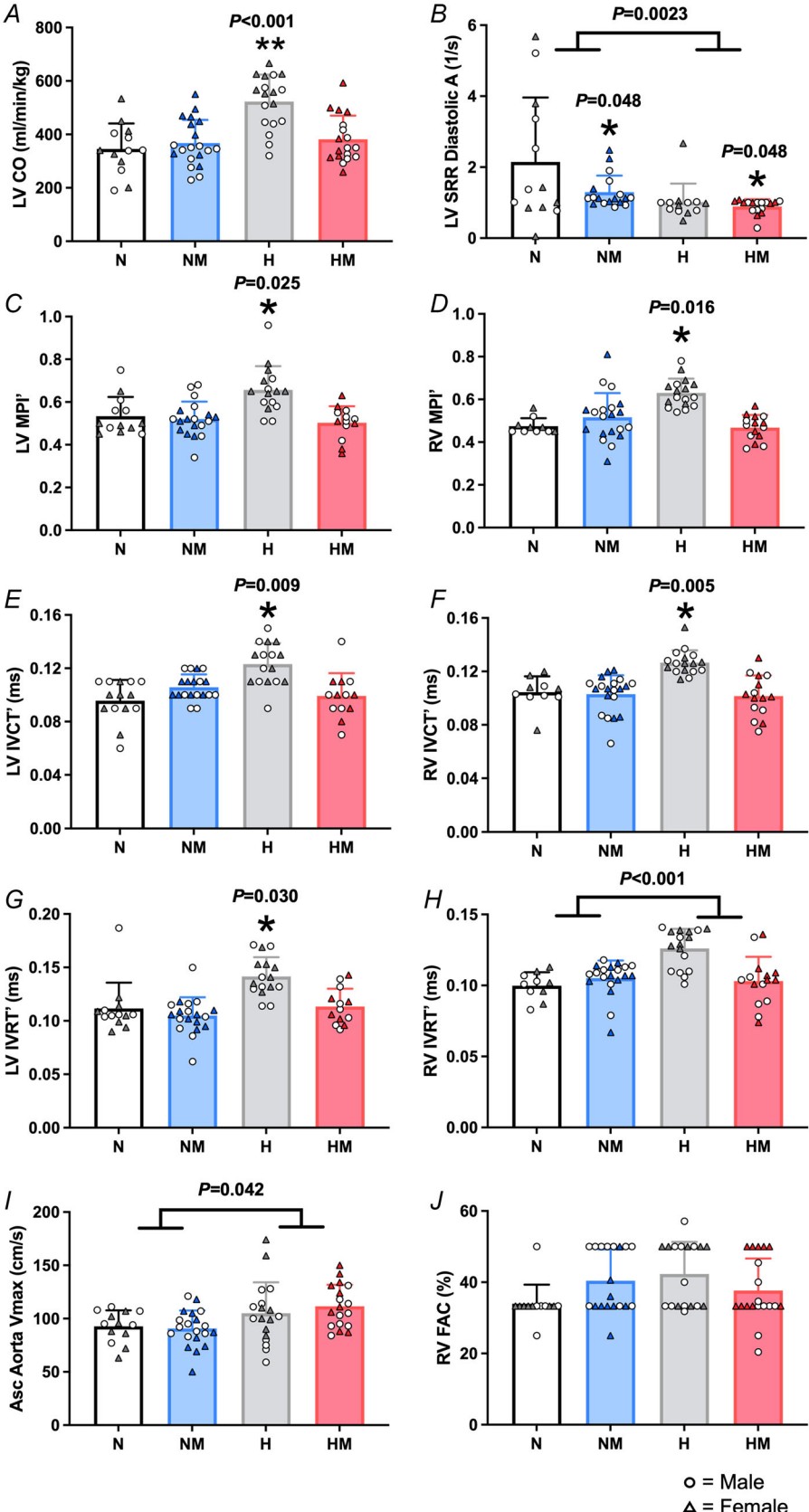

**Figure 1. Effect of chronic hypoxia and maternal melatonin treatment during gestation on cardiovascular function in adult offspring of hypoxic pregnancies**
Values are means ± SD. N, Normoxia (*n* = 13); NM, Normoxia Melatonin (*n* = 20); H, Hypoxia (*n* = 18); HM, Hypoxia Melatonin (*n* = 18). Open circles, male offspring; filled triangles, female offspring. \*$P < 0.05$, \*\*$P < 0.01$. *n* = 69 animals total. *A*, CO, cardiac output. *B*, SRR, strain rate radial. *C* and *D*, MPI′, myocardial performance index in the LV and RV, respectively. *E* and *F*, IVCT′, isovolumetric contraction time in the LV and RV, respectively. *G* and *H*, IVRT′, isovolumetric relaxation time in the LV and RV, respectively. *I*, ascending aorta $V_{max}$. *J*, RV fractional area change (FAC). LV, left ventricle; RV, right ventricle. Myocardial performance index′ = (isovolumetric contraction time′ + isovolumetric relaxation time′)/ejection time′ derived by pulsed wave tissue Doppler technique. Shortening fraction = [(ventricular end-diastolic dimension − end-systolic dimension)/end-diastolic dimension] × 100. Myocardial time intervals were normalized by cardiac cycle length adjusting for variation in heart rate. Data were analysed using a mixed linear model. The main effects of oxygen/treatment/sex were assessed simultaneously. [Colour figure can be viewed at wileyonlinelibrary.com]

offspring was also increased by chronic fetal hypoxia (Table 1; interaction, $P = 0.035$). In contrast, in hearts of adult offspring of hypoxic pregnancy the mitral valve to tricuspid valve dimension ratio (MV/TV ratio) remained unchanged (Table 1).

**Effects of maternal melatonin treatment.** Maternal antenatal melatonin treatment prevented the changes in adult offspring LV wall thickness and AV/PV ratio in hypoxic pregnancies (Table 1). In both normoxic and hypoxic groups, maternal melatonin treatment decreased the MV/TV ratio in adult offspring (Table 1; main effect, $P < 0.001$).

## Cardiomyocyte diastolic function measured with fluorescent microscopy

**Effects of chronic fetal hypoxia.** The diastolic dysfunction measured with echocardiography in offspring from hypoxic pregnancies was associated with abnormal calcium cycling. Chronic fetal hypoxia compromised the kinetics of diastolic calcium cycling in male and female adult offspring, as evidenced by an increased DT50 (Fig. 2 and Table 2; interaction, $P = 0.003$) and a prolonged decay time constant (tau) of the $Ca^{2+}$ transient (Fig. 2 and Fig. 3*C* and *D*; interaction, $P = 0.007$), compared to controls. These changes increased the time taken for ventricular cardiomyocytes to relax to 50% of the amplitude of contraction, but this effect was only significant in the RV (Fig. 3*F*; main effect, $P = 0.042$), and there were no differences in diastolic levels of [$Ca^{2+}$] concentration in either the LV or RV (Fig. 3*A* and *B*). There was no change in diastolic velocities (rate of cellular relaxation) in each ventricle when the total amplitude of cardiomyocyte contraction was considered (Table 2).

**Effects of maternal melatonin treatment.** Maternal antenatal melatonin treatment had no effects on diastolic variables in cardiomyocytes from adult offspring in the normoxic group. However, maternal melatonin treatment prevented the increased $Ca^{2+}$ transient tau and the right ventricular DT50 in hearts of adult offspring from hypoxic pregnancies. However, maternal antenatal melatonin treatment had no effect on the rate of relaxation in the RV, or DT50 in the LV.

## Cardiomyocyte systolic function measured with fluorescent microscopy

**Effects of chronic fetal hypoxia.** Peak amplitude was reduced, indicating lower systolic $Ca^{2+}$ in isolated cardiomyocytes from adult offspring of hypoxic pregnancies in both ventricles (Fig. 4*A* and *B*; main effect: LV, $P = 0.0057$; RV, $P = 0.033$), and $Ca^{2+}$ transient amplitude was reduced in left ventricular myocytes (Fig. 4*C* and *D*; interaction, $P = 0.034$). These changes occurred with a reduction in the amplitude of contraction in right ventricular myocytes, but not left (Fig. 4*E* and *F*; interaction, $P < 0.05$). Time to peak amplitude of the $Ca^{2+}$ transient and the contraction was unchanged by chronic fetal hypoxia in either ventricle of adult offspring (Table 2), but there was an increase in time to 50% systole of left ventricular myocytes (Fig. 4*G* and *H*; main effect, $P = 0.011$). Furthermore, hypoxic pregnancy led to a decrease in the systolic velocity of right ventricular myocytes in adult offspring, when the total amplitude of cardiomyocyte contraction was considered (Fig. 4*I* and *J*; main effect, $P = 0.039$).

**Effects of maternal melatonin treatment.** Maternal antenatal melatonin treatment had no effect on systolic variables in ventricular myocytes in hearts of adult offspring from the normoxic group. It also had no effect on the reduction in peak systolic amplitude of the $Ca^{2+}$ transient, the prolongation of LV myocyte time to 50% contraction and the reduction in the RV myocyte systolic velocity of contraction in hearts of adult offspring from the hypoxic groups. However, the reduction in LV $Ca^{2+}$ transient amplitude and RV myocyte contraction amplitude in hearts of adult offspring from the hypoxic groups was prevented by maternal melatonin treatment.

**Table 2. Isolated cell measures**

| Measure | Left ventricle | | | | Right ventricle | | | |
|---|---|---|---|---|---|---|---|---|
| | Normoxia (n = 13) | Normoxia melatonin (n = 12) | Hypoxia (n = 12) | Hypoxia melatonin (n = 13) | Normoxia (n = 13) | Normoxia melatonin (n = 12) | Hypoxia (n = 12) | Hypoxia melatonin (n = 13) |
| Diastolic cell length (µm) | 118.3 ± 20.5 | 121.0 ± 16.5 | 115.4 ± 17.4 | 114.2 ± 15.8 | 125.7 ± 16.9 | 114.7 ± 15.0 | 113.6 ± 15.2 | 116.3 ± 15.8 |
| Diastolic cell width (µm) | 28.8 ± 8.8 | 28.2 ± 6.6 | 29.1 ± 8.6 | 29.3 ± 8.9 | 30.7 ± 9.4 | 29.1 ± 8.1 | 27.2 ± 6.9 | 27.9 ± 7.6 |
| Diastolic velocity (amplitude/systolic time) | 30.46 ± 23.91 | 31.58 ± 29.20 | 27.80 ± 22.63 | 29.70 ± 25.78 | 29.07 ± 24.06 | 26.26 ± 14.88 | 22.25 ± 16.11 | 29.06 ± 25.05 |
| Diastolic $Ca^{2+}$ DT50 (ms) | 132.5 ± 37.44 | 159.0 ± 30.87 | 183.5 ± 34.99* | 164.3 ± 34.51* | 136.8 ± 39.59 | 164.5 ± 42.79 | 184.4 ± 27.93* | 158.0 ± 34.44 |
| Diastolic $Ca^{2+}$ DT90 (ms) | 393.58 ± 107.48 | 428.26 ± 107.63 | 479.10 ± 117.00 | 438.50 ± 106.49 | 377.19 ± 79.572 | 393.95 ± 128.40 | 477.80 ± 113.26** | 443.82 ± 110.72** |
| Systolic cell width (µm) | 31.5 ± 9.9 | 30.3 ± 7.4 | 29.8 ± 9.5 | 30.8 ± 9.6 | 31.8 ± 9.5 | 29.8 ± 8.3 | 27.9 ± 7.2 | 28.7 ± 7.9 |
| Systolic contraction time to peak (s) | 0.253 ± 0.148 | 0.286 ± 0.191 | 0.276 ± 0.140 | 0.349 ± 0.187 | 0.284 ± 0.184 | 0.304 ± 0.162 | 0.345 ± 0.229 | 0.302 ± 0.201 |
| Systolic $Ca^{2+}$ time to peak (ms) | 70.48 ± 19.21 | 65.09 ± 21.05 | 66.11 ± 23.20 | 60.94 ± 16.09 | 67.34 ± 19.49 | 66.26 ± 17.92 | 62.62 ± 17.98 | 62.74 ± 21.04 |
| Isoprenaline DT50 (ms) | 120.92 ± 40.289 | 125.32 ± 28.788 | 168.06 ± 31.594* | 141.19 ± 29.128* | 146.96 ± 39.099 | 132.55 ± 32.541 | 162.85 ± 25.866 | 134.58 ± 34.564 |
| Isoprenaline systolic $Ca^{2+}$ ($F_{340/380}$ ratio) | 0.2702 ± 0.0311 | 0.2717 ± 0.0571 | 0.2617 ± 0.0555 | 0.2567 ± 0.0582 | 0.2769 ± 0.0407 | 0.2702 ± 0.0481 | 0.2492 ± 0.0688 | 0.2560 ± 0.0509 |
| Isoprenaline $Ca^{2+}$ amplitude ($\triangle F_{340/380}$) | 0.1084 ± 0.0204 | 0.1149 ± 0.0320 | 0.1133 ± 0.0312 | 0.1059 ± 0.0409 | 0.1202 ± 0.0291 | 0.1008 ± 0.0224 | 0.1028 ± 0.0423 | 0.09968 ± 0.0291 |
| Isoprenaline contraction amplitude (AU) | 9.93 ± 4.03 | 9.50 ± 4.03 | 8.97 ± 4.00 | 8.65 ± 4.35 | 9.92 ± 3.63 | 8.80 ± 3.25 | 8.11 ± 3.91 | 8.31 ± 3.28 |
| Isoprenaline time to peak contraction (s) | 0.295 ± 0.192 | 0.354 ± 0.195 | 0.306 ± 0.182 | 0.312 ± 0.215 | 0.343 ± 0.21 | 0.270 ± 0.187 | 0.358 ± 0.185 | 0.341 ± 0.187 |
| Isoprenaline time 50% systole (s) | 0.130 ± 0.105 | 0.157 ± 0.110 | 0.174 ± 0.096 | 0.140 ± 0.074 | 0.159 ± 0.103 | 0.126 ± 0.090 | 0.166 ± 0.075 | 0.154 ± 0.081 |
| Caffeine amplitude ($\triangle F_{340/380}$) | 0.0912 ± 0.0259 | 0.0921 ± 0.0244 | 0.0890 ± 0.0291 | 0.0921 ± 0.0362 | 0.0916 ± 0.0270 | 0.0891 ± 0.0301 | 0.0865 ± 0.0299 | 0.0883 ± 0.0217 |
| Caffeine DT50 (ms) | 2495 ± 1223 | 2313 ± 1032 | 2709 ± 1037 | 3057 ± 938.7 | 2332 ± 1206 | 2584 ± 1128 | 2518 ± 1185 | 3160 ± 935.6 |
| Caffeine DT90 (ms) | 4926 ± 1859 | 5770 ± 2414 | 5540 ± 2097 | 6526 ± 1860 | 5357 ± 2175 | 6225 ± 2717 | 5873 ± 2194 | 6054 ± 1504 |
| Caffeine rate of decay (tau) | 2616 ± 968.2 | 2522 ± 1374 | 2629 ± 1475 | 2972 ± 1347 | 2768 ± 1454 | 3252 ± 1692 | 3259 ± 1663 | 3317 ± 1618 |

Values are means ± SD. Data were analysed using a mixed linear model. The model included nested measurements within an animal; $n$ = 50 animals total. The main effects of oxygen/treatment/sex were assessed simultaneously. Hypoxia Effect: *$P < 0.05$, **$P < 0.01$. Melatonin effect: †$P < 0.05$, ††$P < 0.01$.

## Effects of isoprenaline on cardiomyocyte calcium cycling and contractility

**Effects of chronic fetal hypoxia.** The diastolic $Ca^{2+}$ dysfunction in ventricular myocytes from adult offspring of hypoxic pregnancies was still apparent in the presence of isoprenaline, as evidenced by a prolonged LV slope decay of the calcium transient (Fig. 5A; interaction, $P = 0.0107$), as well as a prolonged LV and RV DT90 (Fig. 5C and D; $P = 0.0031$, $P = 0.0058$),

compared to controls. The prolonged diastolic $Ca^{2+}$ kinetics in the presence of isoprenaline occurred with slower rates of diastolic relaxation in the RV, but not the LV (Fig. 5*F*; $P = 0.0126$). In contrast to diastole, $\beta$-adrenergic stimulation with isoprenaline restored the transient amplitude and peak systolic amplitude to control

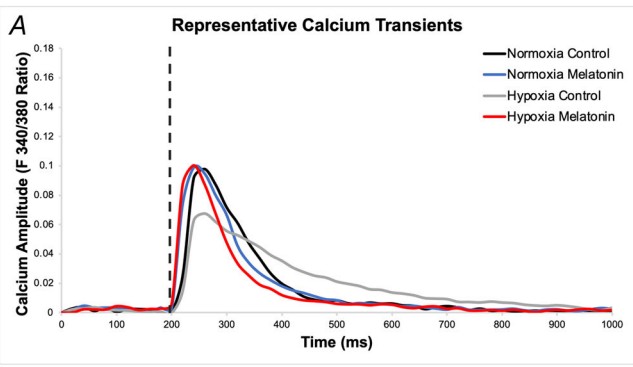

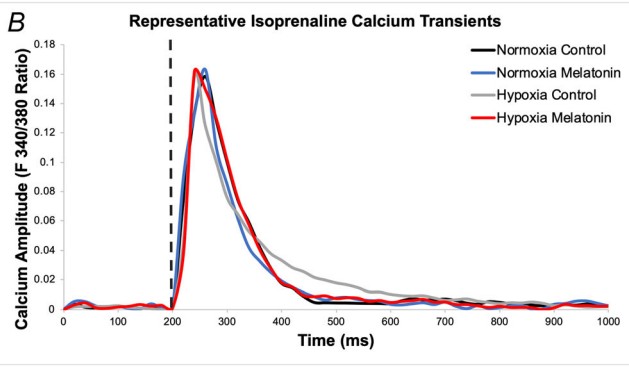

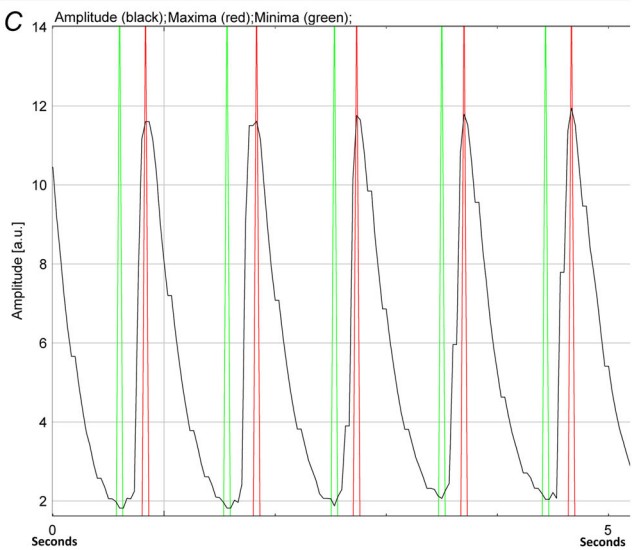

**Figure 2. Representative traces of $Ca^{2+}$ before and after isoprenaline treatment and cardiomyocyte contractile function**
Traces show a representative single $Ca^{2+}$ transient from each treatment group; dashed line represents the start of stimulation. Cardiomyocyte contraction; green, minima; red, maxima. *n*, one representative cardiomyocyte per treatment group. [Colour figure can be viewed at wileyonlinelibrary.com]

levels in ventricular myocytes in adult offspring from the hypoxic group (Table 2). This meant the relative change in $Ca^{2+}$ amplitude after isoprenaline treatment was larger in adult offspring from hypoxic pregnancies, compared to those from normoxic pregnancies (Fig. 5*G*; $P = 0.0114$), indicating a relatively larger response to $\beta$-adrenergic stimulation. Isoprenaline treatment in the hearts of adult offspring in the hypoxic group also restored ventricular myocyte contraction, time to peak contraction and time to 50% systole to normoxic levels (Table 2). However, the velocity of contraction (contraction amplitude/time to systole) of cells isolated from the LV and RV after isoprenaline treatment was lower in the adult offspring from the hypoxic groups compared to those from the normoxic groups (Fig. 5*I* and *J*; main effects, $P = 0.045$, $P = 0.0049$).

**Effects of maternal melatonin treatment.** Maternal antenatal melatonin treatment had no effects on the response to isoprenaline in ventricular myocytes in adult offspring from the normoxic group. It was also unable to prevent the changes in LV $Ca^{2+}$ transient DT90, RV time to 50% of relaxation, and systolic velocity in isoprenaline-treated myocytes in adult offspring from the hypoxic group. However, the changes in LV slope decay of the $Ca^{2+}$ transient, the RV $Ca^{2+}$ transient DT90, and the percentage change in LV $Ca^{2+}$ amplitude of the isoprenaline-treated myocytes in adult offspring from the hypoxic groups were prevented by maternal melatonin treatment.

## Cardiomyocyte structure measured with light-microscopy and video imaging

**Effects of chronic fetal hypoxia.** Despite the changes in ventricular wall thickness and sphericity, there was no difference in cardiomyocyte area (Fig. 6*A–G*), length or width (Table 2) in adult offspring from hypoxic pregnancy.

**Effects of maternal melatonin treatment.** Similarly, maternal antenatal melatonin treatment did not alter isolated cardiomyocyte area (Fig. 6*A–G*), length or width (Table 2) in adult offspring of normoxic or hypoxic pregnancy.

## Discussion

The data reported support the hypothesis that cardiac dysfunction in adult offspring programmed by hypoxic pregnancy is underpinned by maladaptive cardiomyocyte calcium handling. We show that offspring from hypoxic pregnancies have biventricular diastolic dysfunction that is linked to an impaired diastolic recovery of $[Ca^{2+}]_i$. Contractility and the amplitude of $[Ca^{2+}]_i$ were

also reduced in offspring from hypoxic pregnancies, but cardiac output and stroke volume were increased, suggestive of compensatory mechanisms that override the reduced contractility. We also showed offspring from hypoxic pregnancies had a greater increase in systolic $[Ca^{2+}]_i$ amplitude after $\beta$-adrenergic stimulation, indicating cardiac sympathetic sensitization. Lastly, our data suggest maternal oral melatonin treatment can improve developmentally programmed cardiomyocyte dysfunction in hearts of adult offspring of hypoxic pregnancy.

## Cardiac geometry

Cardiac remodelling was evident in adult offspring from hypoxic pregnancy showing an increase in the LV sphericity index, combined with thinner LV walls. Consistent with these findings, similar effects have been reported in humans in the LV of fetuses and of adult offspring from hypoxic growth restricted pregnancies (Crispi et al., 2010; Patey et al., 2019). We have previously reported that this model of hypoxic pregnancy in rats does not programme alterations in basal arterial blood pressure measured via chronically

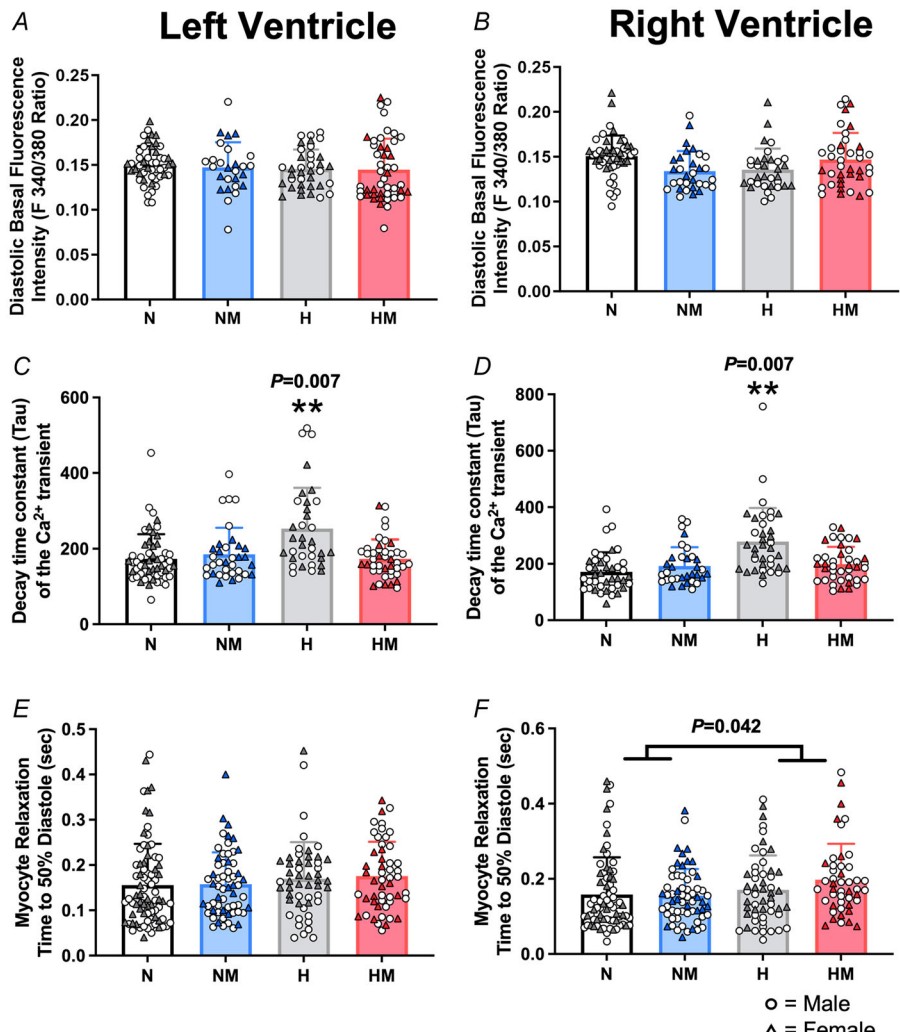

**Figure 3. Diastolic Ca²⁺ measures within isolated cardiomyocytes in the left and right ventricle**
*A* and *B*, diastolic Ca²⁺ measure of resting fluorescence intensity in isolated cardiomyocytes from the left ventricle and right ventricle, respectively. *C* and *D*, time constant for the decay of the Ca²⁺ transient in isolated cardiomyocytes from left and right ventricle, respectively. *E* and *F*, time taken for isolated cells to relax to 50% of the amplitude of contraction from isolated cardiomyocytes from the left and right ventricle, respectively. Values are means ± SD. N, Normoxia (*n* = 13); NM, Normoxia Melatonin (*n* = 12); H, Hypoxia (*n* = 12); HM, Hypoxia Melatonin (*n* = 13). Open circles, male offspring; filled triangles, female offspring. *P < 0.05, **P < 0.01. Data were analysed using a mixed linear model. Each data point represents an individual cell; the model included nested measurements within an animal; *n* = 50 animals total with approximately 10 cells per animal. The main effects of oxygen/treatment/sex were assessed simultaneously. [Colour figure can be viewed at wileyonlinelibrary.com]

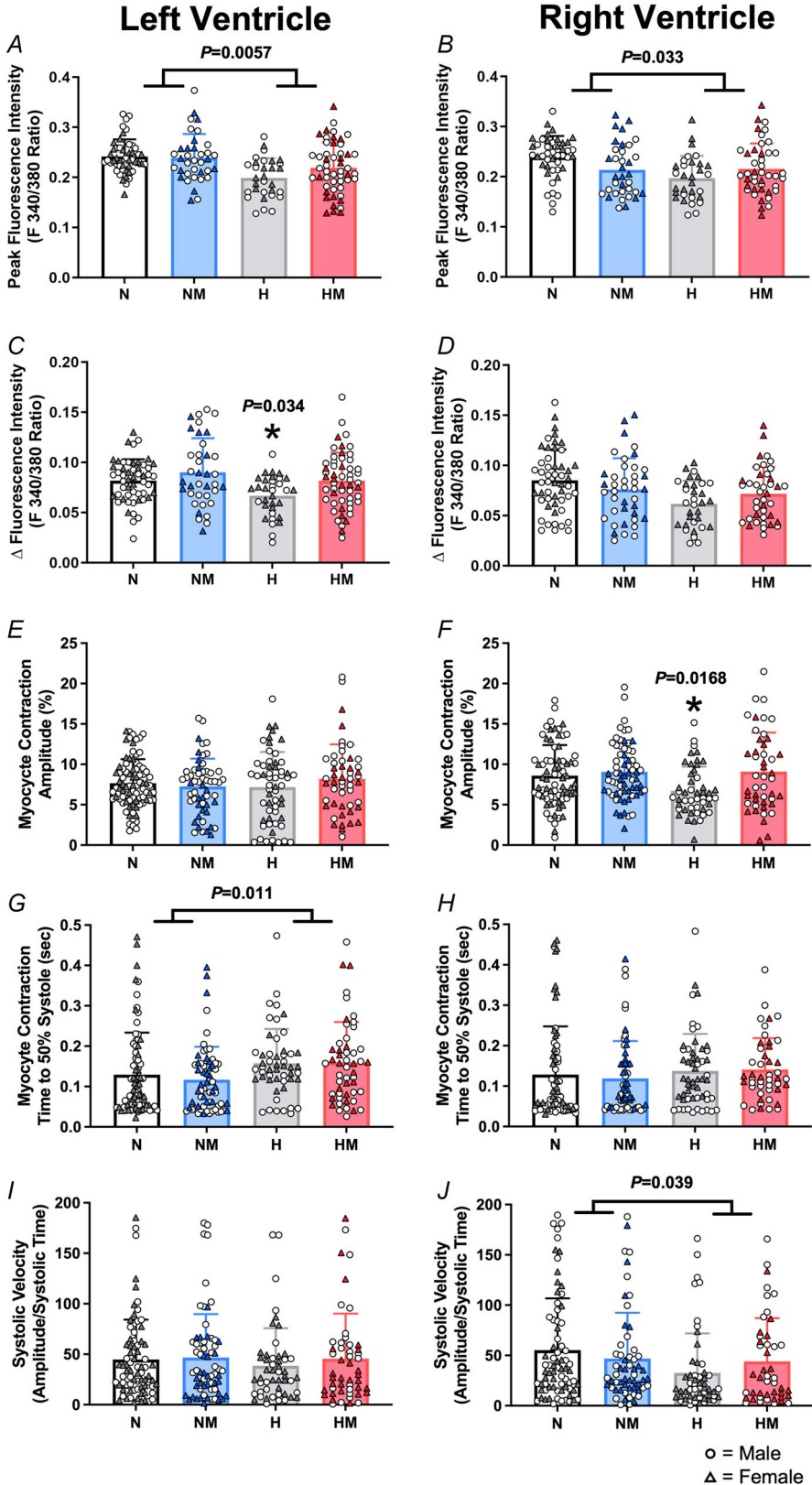

**Figure 4. Systolic Ca²⁺ measures within isolated cells in the left and right ventricle**
*A* and *B*, systolic Ca²⁺ measure of peak fluorescence intensity in isolated cardiomyocytes from the left ventricle and right ventricle, respectively. *C* and *D*, amplitude of Ca²⁺ transients measured as the difference between diastolic

baseline and peak fluorescence intensity in isolated cardiomyocytes from the left ventricle and right ventricle, respectively. *E* and *F*, amplitude of isolated cardiomyocyte contraction from the left and right ventricles, respectively. *G* and *H*, time taken for isolated cardiomyocytes to reach 50% of peak contraction from the left and right ventricles, respectively. *I* and *J*, velocity of contraction (amplitude of contraction divided by time for maximal contraction) of isolated cardiomyocytes from the left and right ventricle, respectively. Values are means ± SD. N, Normoxia ($n = 13$); NM, Normoxia Melatonin ($n = 12$); H, Hypoxia ($n = 12$); HM, Hypoxia Melatonin ($n = 13$). Open circles, male offspring; filled triangles, female offspring. *$P < 0.05$, **$P < 0.01$. Data were analysed using a mixed linear model. Each data point represents an individual cell; the model included nested measurements within an animal; $n = 50$ animals total with approximately 10 cells per animal. The main effects of oxygen/treatment/sex were assessed simultaneously. [Colour figure can be viewed at wileyonlinelibrary.com]

instrumented indwelling catheters in 4-month adult offspring (Lakshman et al., 2021; Spiroski et al., 2021). Therefore, effects of adult offspring ventricular remodelling programmed by developmental hypoxia are independent of effects on the adult offspring cardiac afterload. The ventricular remodelling is therefore more likely to represent developmentally programmed effects triggered by the fetal brain sparing response to hypoxic pregnancy (Giussani, 2016). This can promote sustained changes in fetal biventricular preload and afterload, leading to persisting effects in the structure of the adult heart. Alternatively, cardiac remodelling could develop with age after exposure to chronic fetal hypoxia (Rueda-Clausen et al., 2008). For example, previous work has shown that offspring from hypoxic pregnancies have a thickened pulmonary artery, an indicator of pulmonary hypertension (Ding et al., 2020; Papamatheakis et al., 2013). The increased pulmonary vascular resistance places strain on the RV, and with age, it could eventually trigger RV hypertrophy. Remodelling of the RV often results in the impairment of LV geometry, structure and function. RV remodelling can precipitate LV failure through increased pulmonary pressure, augmented left ventricular workload, impaired filling and systemic compensatory mechanisms (Rosenkranz et al., 2020). This condition often leads to a reduction in LV end-diastolic volume and LV mass, as well as decreased LV systolic strain, stroke volume and ejection fraction (Rosenkranz et al., 2020). Hence at 4 months old, equivalent to early adulthood in humans, the rats in our study appear to have a predisposition to developing compensated followed by decompensated phases of heart failure (Adusumalli & Mazurek, 2017). These data are also consistent with our finding of an increase in cardiac sphericity index in adult offspring of hypoxic pregnancy. The latter was due to the combined effects of an increase in LVEDD and a decrease in end diastolic length, although either variable alone did not reach significance. Interestingly, the effects of hypoxic pregnancy on ventricular wall thickness were not accompanied by any change in cardiomyocyte cell size in either ventricle in this study. This result indicates that the RV wall thickening and the LV wall thinning were driven by either; changes in cardiomyocyte number rather than size or a change in cellular composition, for example an

increased proportion of cardiac fibroblast population within the heart of offspring of hypoxic pregnancy.

## Cardiac function

In the present study, offspring from hypoxic pregnancy had an increased MPI′ in both the LV and RV which was driven by a prolonged IVCT′ and IVRT′. Increased MPI′ is a recognized sensitive measure of impaired global myocardial performance (Goroshi & Chand, 2016). Consistent with a prolonged IVRT′, offspring of hypoxic pregnancy had a slower rate of decay of the $[Ca^{2+}]_i$ transient, evidenced by a significant prolongation of LV DT50 and tau, as well as RV DT90 and tau. This may have been driven by several mechanisms, including a reduced rate of $[Ca^{2+}]_i$ reuptake into the sarcoplasmic reticulum, which is a common driver of diastolic dysfunction and one of the many hypothesized underlying mechanisms leading to heart failure (Briston et al., 2011; Eisner et al., 2020). The prolongation of IVCT′ was also associated with a longer time to 50% systole in LV myocytes, but not RV myocytes. Lastly, the changes in LV function occurred with a reduction in radial strain rate, again indicating increased LV afterload promoting a more globular cardiac phenotype.

The data in the present study also show that offspring from hypoxic pregnancies had an increase in stroke volume and CO, while heart rate was unchanged. This is consistent with the additional echocardiography data presented showing a significant increase in aortic valve dimension and aortic velocity time integral, reflecting an increase in CO. However, the isolated myocyte, contractility was reduced in the RV and LV, and this was associated with a reduction in systolic $[Ca^{2+}]_i$ amplitude. These results suggest compensatory mechanisms are triggered *in vivo* to offset the reduced cardiomyocyte contractility and maintain CO. For instance, offspring from hypoxic pregnancies are known to have increased sympathetic activity (Galli et al., 2022), which helps maintain cardiac output under conditions of enhanced afterload in adult offspring of hypoxic pregnancy. Consistent with these findings, we found that $\beta$-adrenergic stimulation of cardiomyocytes in offspring

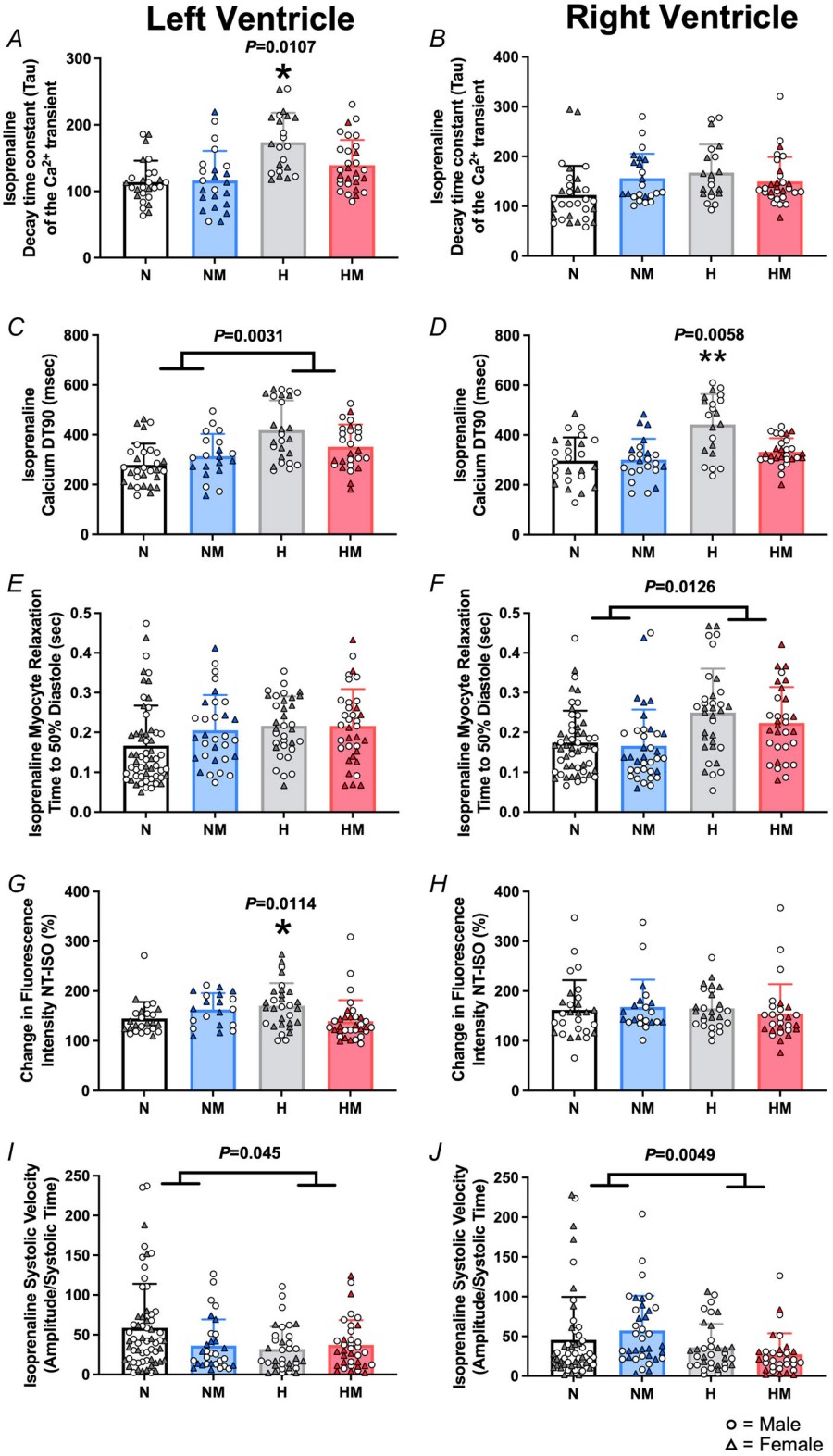

**Figure 5. Diastolic Ca$^{2+}$ and contraction measures within isolated cells after isoprenaline in the left and right ventricle**

*A* and *B*, time constant for the decay of the Ca$^{2+}$ transient in isolated cardiomyocytes after isoprenaline treatment from left and right ventricle respectively. *C* and *D*, time taken for 90% reduction in Ca$^{2+}$ from peak in cellular

diastole from isolated cardiomyocytes after isoprenaline treatment from the left and right ventricle respectively. *E* and *F*, time taken for isolated cells to relax to 50% of the amplitude of contraction from isolated cardiomyocytes after isoprenaline treatment from the left and right ventricle, respectively. *G* and *H*, change in $Ca^{2+}$ amplitude from normal Tyrode (NT) to isoprenaline treatment measured as a percentage in isolated cardiomyocytes from the left and right ventricle, respectively. *I* and *J*, velocity of contraction (amplitude of contraction divided by time for maximal contraction) of isolated cardiomyocytes after isoprenaline treatment from the left and right ventricle, respectively. Values are means ± SD. N, Normoxia ($n = 13$); NM, Normoxia Melatonin ($n = 12$); H, Hypoxia ($n = 12$); HM, Hypoxia Melatonin ($n = 13$). Open circles, male offspring; filled triangles, female offspring. *$P < 0.05$, **$P < 0.01$. Data were analysed using a mixed linear model. Each data point represents an individual cell, the model included nested measurements within an animal; $n = 50$ animals with approximately 5 cells per animal. The main effects of oxygen/treatment/sex were assessed simultaneously. [Colour figure can be viewed at wileyonlinelibrary.com]

from hypoxic pregnancy normalized the systolic $[Ca^{2+}]_i$ amplitude, compared to normoxic controls. This meant a relatively larger change in systolic $[Ca^{2+}]_i$ amplitude, indicating a sensitized response to adrenergic stimulation. Despite $\beta$-adrenergic stimulation and the normalization of the systolic $Ca^{2+}$ transient, there was still prominent diastolic dysfunction present in cardiomyocytes isolated from offspring of hypoxic pregnancy. While enhanced sympathetic drive may help to maintain cardiac output when contractility is compromised, it is unsustainable and eventually becomes a hallmark of early-stage heart failure (Bristow, 2000; Danson et al., 2009). Therefore, combined, the data presented in this study support that gestational hypoxia programmes maladaptive changes in calcium handling in the cardiomyocyte contributing to early stages of heart failure in male and female adult offspring of hypoxic pregnancy.

### Intervention with antenatal melatonin treatment

Additional data in the present study show that maternal antenatal melatonin treatment was effective in correcting many of the adverse ventricular outcomes in male and female offspring from hypoxic pregnancies. Indeed, maternal melatonin treatment during hypoxic pregnancy prevented aortic valve remodelling and reduced the AV/PV size ratio in the adult offspring. Importantly, melatonin was also successful in reducing the MPI′, CO, IVCT′ and IVRT′ of both ventricles, as well as restoring cardiomyocyte systolic $[Ca^{2+}]_i$ amplitude, contractility and velocity of contraction. Maternal antenatal treatment with melatonin also reverted several cellular measures of diastolic function by reducing time constant (tau) of the $Ca^{2+}$ transient in the LV and improving DT90 in the RV of animals exposed to hypoxic pregnancy. Therefore, this work extends previous findings by our group in ovine, rodent and avian preclinical models, which showed the potential for melatonin treatment to prevent oxidative stress directly in the developing heart of chronically hypoxic embryos (Itani et al., 2016), and to protect susceptibility to cardiovascular dysfunction of adult offspring of hypoxic pregnancy (Hansell et al., 2022). Importantly, the data in the present paper show that antenatal beneficial effects of melatonin on the

cardiovascular system of progeny transcend through to protection against developmentally programmed cardiac dysfunction at the level of calcium handling within the cardiomyocyte of adult offspring. Since the mother in this rodent experimental model is exposed to hypoxia, it could be argued that some of the effects of hypoxic pregnancy programming an increased risk of cardiac dysfunction in the adult offspring may be due to maternal effects in addition to direct effects of chronic hypoxia on the developing fetal cardiovascular system. We have previously reported that this model of hypoxic pregnancy in rats is sufficient to lead to an increase in maternal haematocrit (Giussani et al., 2012) and the subtle induction of placental molecular indices of oxidative stress by the end of pregnancy (Richter et al., 2012). Indeed, maternal treatment with other antioxidants, such as vitamin C, in this model of maternal hypoxia protects against enhanced placental oxidative stress (Richter et al., 2012). Melatonin has potent antioxidant effects, which have been shown to protect against placental oxidative stress in other models of compromised pregnancy in rats (Richter et al., 2009); it is being trialled clinically for protecting against fetal brain damage and low birth weight in human pregnancy affected by fetal growth restriction (Palmer et al., 2019); and it is also known to promote an increase in umbilical blood flow by quenching excess free radical production and increasing the bioavailability of nitric oxide (Thakor et al., 2010). Therefore, some of the protective effects on programmed cardiac dysfunction in the adult offspring of maternal treatment with melatonin in hypoxic pregnancy in this model may be due to similar effects at the level of the placenta, protecting uteroplacental perfusion. Future integrative programmes of work, outside the scope of the present study, will address this possibility, combining studies of maternal cardiovascular function, uteroplacental vascular reactivity and placental molecular biology. Finally, because maternal treatment with melatonin in normoxic pregnancy caused a reduction in the offspring MV/TV ratio in the present study, caution should be applied when treating mothers prophylactically. Given the results of our study, clinical trials should consider further investigation into the dosage, timing and appropriateness of melatonin use in all mothers, and that antenatal melatonin therapy may

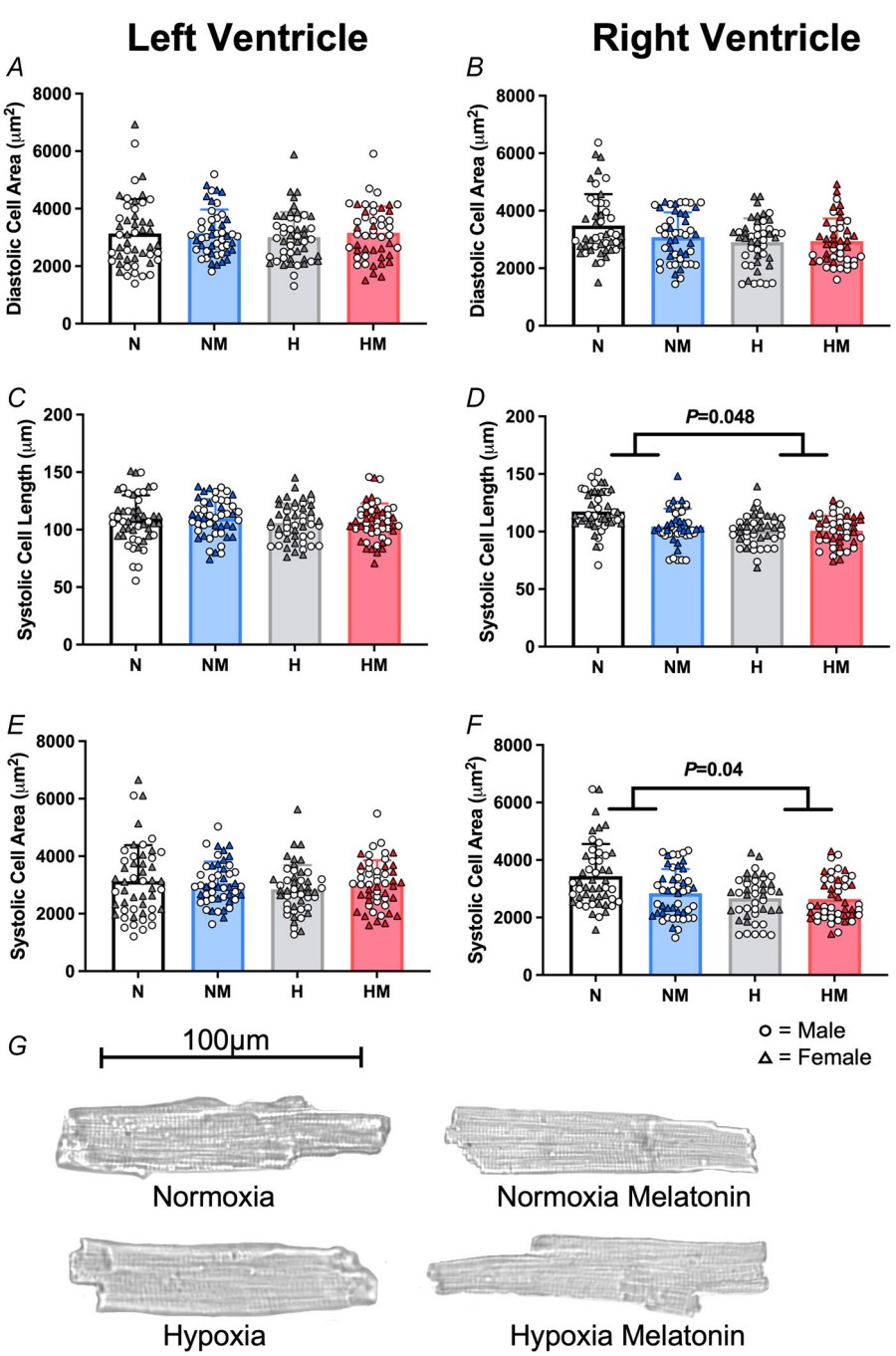

**Figure 6. Systolic isolated cell size measures and example cell micrographs**
*A* and *B*, diastolic cell area of isolated cardiomyocytes from the left and right ventricle, respectively. *C* and *D*, systolic cell length of isolated cardiomyocytes from the left and right ventricle, respectively. *E* and *F*, systolic cell area of isolated cardiomyocytes from the left and right ventricle, respectively. *G*, representative example cardiomyocytes from N (top left), NM (top right) and H (bottom left) and HM (bottom right) treatment groups. Values are means $\pm$ SD. N, Normoxia ($n$ = 13); NM, Normoxia Melatonin ($n$ = 12); H, Hypoxia ($n$ = 12); HM, Hypoxia Melatonin ($n$ = 13). Open circles, male offspring; filled triangles, female offspring. $*P < 0.05$, $**P < 0.01$. Data were analysed using a mixed linear model. Each data point represents an individual cell; the model included nested measurements within an animal; $n$ = 50 animals with approximately 10 cells per animal. The main effects of oxygen/treatment/sex were assessed simultaneously. [Colour figure can be viewed at wileyonlinelibrary.com]

only be appropriate for use once a diagnosis of hypoxic pregnancy has been made.

## Strengths and limitations

An asset of the present study is that it linked echocardiographic data with cardiomyocyte calcium homeostasis to reveal a cellular mechanism for cardiac dysfunction in rat offspring from hypoxic pregnancy. The isolated cell approach allowed us to investigate how the functional units of the heart were affected in the absence of systemic influences such as changes in vascular tone, sympathetic innervation and other factors known to be altered as a result of hypoxic pregnancy. In addition, the study design involved measurement of all outcomes in male and female adult offspring in all groups, using only one rat per litter per outcome variable measured, thereby negating possible confounding effects introduced by within-litter variation. Mixed linear model analysis only detected minor sex-dependent effects within our dataset. Therefore, the study highlights that programmed heart disease by gestational hypoxia and antenatal protection by maternal treatment with melatonin is just as important for women as it is for men.

A limitation of this study is that we were not able to assess the function of ion channels in this cohort. We can infer that there was no change in NCX and PMCA function as there was no difference in the recovery of $Ca^{2+}$ after caffeine-induced calcium release in isolated cells. It is, however, possible that changes in other ion channels may have occurred leading to altered $Ca^{2+}$ buffering. While this study is the first to identify underlying changes in $Ca^{2+}$ handling as a possible causative mechanism for the increased incidence of heart disease in adult offspring of hypoxic pregnancy, further studies are required to determine possible alterations in the expression of proteins within the excitation–contraction coupling pathway.

## Conclusion

This study identified important changes in $Ca^{2+}$ handling within cardiomyocytes isolated from offspring of hypoxic pregnancy including reduced peak amplitude, indicating lower systolic $Ca^{2+}$ transients, impaired diastolic recovery of $[Ca^{2+}]_i$ and a greater increase in systolic $[Ca^{2+}]_i$ amplitude to $\beta$-adrenergic stimulation. These changes in cardiomyocyte $Ca^{2+}$ handling help to explain dysregulation of the biventricular systolic and diastolic dysfunction determined by echocardiography. In addition, we established the efficacy of antenatal antioxidant treatment in the form of maternal oral melatonin to prevent the cardiomyocyte dysfunction in the adult offspring programmed developmentally by hypoxic pregnancy. The data show protection against programmed maladaptive cardiomyocyte calcium handling and thereby improvement in cardiac function in adult offspring of hypoxic pregnancy whose mothers were treated with melatonin with doses lower than those recommended for overcoming jet lag in humans. This supports the use of maternal antenatal treatment with melatonin as effective therapy with good human translational potential for protection against programmed heart disease in offspring of hypoxic pregnancy. However, melatonin treatment alone in normoxic pregnancy did cause some alterations in cardiac structure in the adult offspring. Therefore, maternal treatment with melatonin should only be given to pregnancies which need it, that is, in pregnancy affected by chronic fetal hypoxia rather than prophylactically to all pregnancies.

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

## Additional information

### Data availability statement

All data supporting the results are presented in the manuscript.

### Competing interests

The authors have no conflict of interest to disclose.

### Author contributions

M.C.L., O.V.P., A.W.T., D.A.G. and G.L.J.G. were responsible for the conception and design of the experiments. M.C.L., O.V.P., K.L.M.S., Y.N. and S.G. performed experiments. M.C.L., O.V.P., D.A.G. and G.G. analysed the data. M.C.L., O.V.P., D.A.G. and G.L.J.G. drafted the paper. All authors edited the paper. G.L.J.G., A.W.T. and D.A.G. obtained funding. All authors have read and approved the final version of this manuscript and agree to be accountable for all aspects of the work in ensuring that questions related to the accuracy or integrity of any part of the work are appropriately investigated and resolved. All persons designated as authors qualify for authorship, and all those who qualify for authorship are listed.

### Funding

This study was funded by a British Heart Foundation (BHF) project grant awarded to G.G., A.T. and D.G. (PG/18/5/33 527).

### Acknowledgements

The authors would like to thank the staff of the Biological Services Facility, University of Manchester. and the staff of the University Biomedical Services, University of Cambridge.

### Keywords

antioxidants, calcium handling, cardiomyocyte, fetus, hypoxia, pregnancy

## Supporting information

Additional supporting information can be found online in the Supporting Information section at the end of the HTML view of the article. Supporting information files available:

**Peer Review History**

