## [Peer Review History · The Journal of Physiology]

Maladaptive Cardiomyocyte Calcium Handling in Adult Offspring of Hypoxic Pregnancy: Protection by Antenatal Maternal Melatonin

Mitchell C Lock, Olga Patey, Kerri LM Smith, Youguo Niu, Ben Jaggs, Andrew W Trafford, Dino A Giussani, and Gina Lucia Jane Galli

DOI: 10.1113/JP287325

Corresponding author(s): Gina Galli (gina.galli@manchester.ac.uk)

The following individual(s) involved in review of this submission have agreed to reveal their identity: Matthew Gorr (Referee #1)

Review Timeline:

Submission Date:	17-Jul-2024
Editorial Decision:	30-Aug-2024
Revision Received:	17-Sep-2024
Accepted:	10-Oct-2024

Senior Editor: Laura Bennet

Reviewing Editor: Christopher Lear

Transaction Report:

Dear Dr Lock,

Re: JP-RP-2024-287325 "Maladaptive Cardiomyocyte Calcium Handling in Adult Offspring of Hypoxic Pregnancy: Protection by Antenatal Maternal Melatonin" by Mitchell C Lock, Olga Patey, Kerri LM Smith, Youguo Niu, Ben Jaggs, Andrew W Trafford, Dino A Giussani, and Gina Lucia Jane Galli

Thank you for submitting your manuscript to The Journal of Physiology. It has been assessed by a Reviewing Editor and by 2 expert referees and we are pleased to tell you that it is acceptable for publication following satisfactory revision.

REVISION CHECKLIST:

Please upload two versions of your manuscript text: one with all relevant changes highlighted and one clean version with no changes tracked. The manuscript file should include all tables and figure legends, but each figure/graph should be uploaded as separate, high-resolution files. The journal is now integrated with Wiley's Image Checking service. For further details, see: <https://www.wiley.com/en-us/network/publishing/research-publishing/trending-stories/upholding-image-integrity-wileys->

image-screening-service

We look forward to receiving your revised submission.

Yours sincerely,

Laura Bennet
Senior Editor
The Journal of Physiology

REQUIRED ITEMS

- Method of euthanasia: please clarify/specify in the Methods section that a rising concentration of CO₂ was used.
- Author photo and profile. First or joint first authors are asked to provide a short biography (no more than 100 words for one author or 150 words in total for joint first authors) and a portrait photograph. These should be uploaded and clearly labelled together in a Word document with the revised version of the manuscript. See Information for Authors for further details.

EDITOR COMMENTS

Reviewing Editor:

Thank you for submitting your interesting study to The Journal of Physiology. The reviewers have been overall positive about the study's importance and rigor, but have asked for some streamlining and clarification throughout. It would also be appropriate to expand the mechanistic speculation in the discussion. Please address the reviewers concerns and comments, in addition to my further comments below.

Specific comments

Introduction

Line 63 - references in different format

Line 63-69 - please clarify if these studies are human/animal and the age of follow up

Line 70 - clarifying animals used by studies would likely be helpful here. If it is multiple species and clumsy to list all I am happy for this comment to be ignored.

Line 74 - again I think species and age of ischaemia would be useful

Line 77 - age of follow up please

Line 85 - suggest biventricular dysfunction rather than failure considering the clinical connotations of 'failure'

I agree with reviewer 1 that the section on excitation-contraction coupling can be reduced significantly, similarly the introduction to beta adrenergic effects could be streamlined

Line 108 - "is any species"

Line 108 - "This is important as data..." this sentence is a bit redundant

Line 118 - please clarify organs investigated in these studies

Line - "By prophylactically reducing oxidative stress for the fetus, offspring adult health can thereby be improved." This is redundant

Methods

Line 178 - the methods detail plasma melatonin concentration measurement but this data is not provided. It is also not detailed when the plasma samples were taken. Further details with results are needed, or remove from methodology and discuss with reference to prior studies

Results

Beginning the results section with sex-dependent effects is somewhat jarring. I suggest it would be best to briefly state that the majority of results did not reveal a sex-dependent effect and thus the majority of data has been combined from both sexes, and then discuss the sex dependent effects with the related results, rather than all at the start.

Discussion

Line 393 - it would be useful to more explicitly detail the authors' proposed connection between the altered Ca²⁺ transients and the observed cardiac dysfunction. In particular the connection between reduced transient amplitude and impaired diastolic recovery with a compensatory increase in CO is unclear and needs further explanation.

Line 423 - I am not convinced that deeming these changes to represent (a state worse than) 'compensated heart failure' is correct. Please reconsider whether this model has generated 'heart failure' vs. early changes that would predispose to a higher chance of developing HF. A study detailing the late adulthood/elderly phenotype of this model would really be needed to demonstrate how many subjects develop HF without superimposed trigger eg ischaemia.

Line 427 - 'significant changes' vs 'significance'

REFEREE COMMENTS

Referee #1:

In this manuscript, the authors demonstrate that in utero hypoxia causes a reduction in cardiac function in offspring, evidenced by state-of-the-art methods for measurement of organ and cardiac myocyte function. The work is very well-presented and statistically rigorous. The use of melatonin to alleviate the observed affect is striking and of possible significant impact in the field. I have a few comments to help with the clarity of this work:

Much of the introduction could be reduced, as there is no need to describe excitation-contraction coupling, for example.

Please indicate that the calcium transients were baseline-normalized to for each sample group, and if any background correction was utilized.

Cardiomyocytes were paced at 1 Hz, which is well below the normal heart rate of a rat in vivo . Please indicate the rationale for this.

As calibration for determination of Ca²⁺ was not done (i.e. the y-axis in figure 4 is not [Ca²⁺], but fluorescence), please change the language in the results to indicate the actual measure used, and not the calcium concentration. For example, it would be more accurate to state "peak amplitude was decreased, indicating lower systolic calcium," rather than "Systolic Ca²⁺ was reduced." Additionally, the y-axis in figure 2 should be changed to F 340/380 or Fura 2 340/380.

The figure legend should indicate the n/group, not the total n. n=10-12 animals per group is sufficient.

Representative myocytes in Fig.6 should be labeled in the figure image.

Line 414: The thickened pulmonary artery does not necessarily promote PH, but is an indicator.

Please describe how, in this model, there may be secondary effects as the mother is also receiving reduced oxygen.

I am not sure I agree with the statement "between compensated and decompensated phases of heart failure." The degree of hypertrophy or other measures demonstrated here do not indicate remodeling close to actual models of heart failure, nor is it demonstrated that the rats would continue towards decompensation given the right stress (time or injury, for example). This is also true of the statement that MPI' is "used clinically to diagnose cardiovascular disorders including dilated cardiomyopathy, ischemic heart disease and heart failure..." The authors should consider removing this statement as it implies that a statistical reduction in one measure is indicative of a disease state.

The statement "Clinically, this finding suggests not all mothers should be treated, but antenatal melatonin therapy should only be applied once a diagnosis of hypoxic pregnancy has been made" should be removed, as this work does not demonstrate clinical outcomes nor consider the timing of administration.

Referee #2:

This is an interesting study examining the impact of maternal hypoxia during most the pregnancy period (days 6 to 20) on adult offspring cardiac function. Both systolic and diastolic function were evaluated using echocardiography and isolated cardiomyocytes were used to assess Ca²⁺ transients.

The main findings corroborate previous studies by the authors that hypoxia during pregnancy induces long-lasting effects on the offspring cardiovascular. The authors also show in the present study that treatment of the mothers during pregnancy with melatonin prevented (or at least attenuated) most of the alterations observed in cardiac function in vivo and in isolated cardiomyocytes. Lastly, modest sex differences were observed when comparing male and female offspring.

Comments:

1) Lines 393 to 396: the authors state that the changes observed in Ca²⁺ handling are consistent with systolic and diastolic

dysfunction assessed by echocardiography, including increased cardiac output. Why would cardiac output be increased by any of the alterations observed in "cardiac" Ca²⁺ in offspring from hypoxic pregnancies?

Cardiac output is not determined by the heart, unless the heart is in failure. Thus, the heart can, under certain circumstances limit cardiac output, but would never cause sustained increases in cardiac output.

2) The discussion about potential mechanisms responsible for the beneficial effect of melatonin during pregnancy to protect against the deleterious effects of hypoxia is very modest and basically concentrates on melatonin "antioxidant" actions. The Discussion section would benefit from a more broad and speculative discussion on potential mechanisms in addition to anti-oxidative properties on melatonin. Is there strong evidence that offspring from hypoxic pregnancies show increased ROS generation or reduced antioxidant capacity? These parameters, perhaps, could have been measured in isolated cardiomyocytes (although, this reviewer is not suggesting that this should be included in the present study).

3) Was blood pressure measured in the offspring? The major cause of systolic and diastolic dysfunction in humans is elevated blood pressure.

END OF COMMENTS

Confidential Review

17-Jul-2024

EDITOR COMMENTS

Reviewing Editor:

Thank you for submitting your interesting study to The Journal of Physiology. The reviewers have been overall positive about the study's importance and rigor, but have asked for some streamlining and clarification throughout. It would also be appropriate to expand the mechanistic speculation in the discussion. Please address the reviewers concerns and comments, in addition to my further comments below.

Specific comments

Introduction

Line 63 - references in different format

We apologise for this referencing issue. The formatting of all references are now in the Journal of Physiology style.

Line 63-69 - please clarify if these studies are human/animal and the age of follow up
Line 70 - clarifying animals used by studies would likely be helpful here. If it is multiple species and clumsy to list all I am happy for this comment to be ignored.

The species for these studies has been added in the manuscript text.

Line 64-74: Importantly, in rats, offspring from hypoxic pregnancies continue to display cardiac abnormalities in adulthood, including left ventricular (LV) wall thickening (Rueda-Clausen et al., 2008) and LV diastolic dysfunction (Rueda-Clausen et al., 2008; Aljunaidy et al., 2018; Niu et al., 2018; Spiroski et al., 2021). In vivo LV systolic function appears to be preserved in offspring from hypoxic rat pregnancies (Rueda-Clausen et al., 2008), resulting from an increase in sympathetic activity and LV contractility, maintaining cardiac output (Hauton & Ousley, 2009; Niu et al., 2018). Comparative measurements in the right ventricle (RV) are lacking, but several studies in rats, chicken and sheep suggest offspring from hypoxic pregnancies develop signs of pulmonary hypertension, including pulmonary arterial wall thickening, decreased pulmonary artery acceleration time and increased RV diameter in diastole (Rueda-Clausen et al., 2008; Botting et al., 2020; Ding et al., 2020; Skeffington et al., 2020; Li et al., 2021).

Line 74 - again I think species and age of ischaemia would be useful.

Species and age of adult offspring has been added to the manuscript.

Line 75: Chronic fetal hypoxia also sensitises the rat heart to ischaemia-reperfusion injury at 3 months of age (Xue & Zhang, 2009), which may predispose to myocardial infarction.

Line 77 - age of follow up please

The age of the study participants has been added.

Line 79: Reported abnormalities in cardiac morphology and function in FGR human fetuses include ventricular wall hypertrophy at 17-24 weeks (Veille et al., 1993), compromised ejection force at 18-38 weeks (Rizzo et al., 1995) and impaired diastolic filling (Miyague et al., 1997). In addition, human clinical studies have reported that low birth weight is associated with globular cardiac ventricles with impaired stroke volume in children (~5 years old), and endothelial dysfunction and systemic hypertension by young adulthood (20-28 years old) (Leeson et al., 2001; Crispi et al., 2010; Bjarnegård et al., 2013).

Line 85 - suggest biventricular dysfunction rather than failure considering the clinical connotations of 'failure'

'failure' has been changed to 'dysfunction', as suggested.

Line 86: Taken together, these data suggest that adult offspring from hypoxic pregnancies have early signs of biventricular dysfunction, increasing cardiovascular risk.

I agree with reviewer 1 that the section on excitation-contraction coupling can be reduced significantly, similarly the introduction to beta adrenergic effects could be streamlined

These sections have been shortened to improve clarity, as requested.

Line 108 - "is any species"

This spelling error has been corrected.

Line 108 - "This is important as data..." this sentence is a bit redundant

The sentence has been removed.

Line 118 - please clarify organs investigated in these studies

This has now been clarified.

Line 113: Importantly, we have recently demonstrated the effectiveness of maternal treatment with melatonin in preventing cardiac oxidative stress in rodent hypoxic pregnancy and there are ongoing human clinical trials for use of melatonin for antioxidant protection in high-risk pregnancies (Hobson et al., 2018; Hansell et al., 2022).

Line - "By prophylactically reducing oxidative stress for the fetus, offspring adult health can thereby be improved." This is redundant

This sentence has been removed.

Methods

Line 178 - the methods detail plasma melatonin concentration measurement but this data is

not provided. It is also not detailed when the plasma samples were taken. Further details with results are needed, or remove from methodology and discuss with reference to prior studies.

Plasma melatonin was published in the previous cited paper. The methods have been removed from this manuscript to avoid confusion.

Results

Beginning the results section with sex-dependent effects is somewhat jarring. I suggest it would be best to briefly state that the majority of results did not reveal a sex-dependent effect and thus the majority of data has been combined from both sexes, and then discuss the sex dependent effects with the related results, rather than all at the start.

The text has been restructured to state that there were few sex effects upfront, as suggested.

Line 289: The majority of the results did not reveal a sex dependent effect; thus, a majority of outcomes were combined and the sex of the rats is denoted in the graphs by different symbols (Fig. 1, 3-6).

Discussion

Line 393 - it would be useful to more explicitly detail the authors' proposed connection between the altered Ca²⁺ transients and the observed cardiac dysfunction. In particular the connection between reduced transient amplitude and impaired diastolic recovery with a compensatory increase in CO is unclear and needs further explanation.

The discussion on the connection between altered calcium transients and observed cardiovascular dysfunction has been expanded to improve clarity of the proposed mechanism.

Line 429: The data in the present study also show that offspring from hypoxic pregnancies had an increase in stroke volume and CO, while heart rate was unchanged. This is consistent with the additional echocardiography data presented showing a significant increase in aortic valve dimension and aortic velocity time integral, reflecting an increase in CO. However, at the isolated myocyte, contractility was reduced in the RV and LV, and this was associated with a reduction in systolic [Ca²⁺]_i amplitude. These results suggest compensatory mechanisms are triggered in vivo to offset the reduced cardiomyocyte contractility and maintain CO. For instance, offspring from hypoxic pregnancies are known to have increased sympathetic activity (Galli et al., 2022), which helps maintain cardiac output under conditions of enhanced afterload in adult offspring of hypoxic pregnancy. Consistent with these findings, we found that β-adrenergic stimulation of cardiomyocytes in offspring from hypoxic pregnancy normalised the systolic [Ca²⁺]_i amplitude, compared to normoxic controls. This meant a relatively larger change in systolic [Ca²⁺]_i amplitude, indicating a sensitised response to adrenergic stimulation. Despite β-adrenergic stimulation and the normalisation of the systolic Ca²⁺ transient, there was still prominent diastolic dysfunction present in cardiomyocytes isolated from offspring of hypoxic pregnancy. While enhanced sympathetic drive may help to maintain cardiac

output when contractility is compromised, it is unsustainable and eventually becomes a hallmark of early-stage heart failure (Bristow, 2000; Danson et al., 2009).

Line 423 - I am not convinced that deeming these changes to represent (a state worse than) 'compensated heart failure' is correct. Please reconsider whether this model has generated 'heart failure' vs. early changes that would predispose to a higher chance of developing HF. A study detailing the late adulthood/elderly phenotype of this model would really be needed to demonstrate how many subjects develop HF without superimposed trigger eg ischaemia.

We thank the Editor for this suggestion and have softened the tone in this section to reflect a predisposition of disease rather than heart failure itself.

Line 488: Hence at 4 months old, equivalent to early adulthood in humans, the rats in our study appear to have a predisposition to developing compensated followed by decompensated phases of heart failure (Adusumalli & Mazurek, 2017).

Line 427 - 'significant changes' vs 'significance'

The text has now been changed.

REFEREE COMMENTS

Referee #1:

In this manuscript, the authors demonstrate that in utero hypoxia causes a reduction in cardiac function in offspring, evidenced by state-of-the-art methods for measurement of organ and cardiac myocyte function. The work is very well-presented and statistically rigorous. The use of melatonin to alleviate the observed affect is striking and of possible significant impact in the field. I have a few comments to help with the clarity of this work:

Much of the introduction could be reduced, as there is no need to describe excitation-contraction coupling, for example.

These sections have been shortened to improve clarity.

Please indicate that the calcium transients were baseline-normalized to for each sample group, and if any background correction was utilized.

Thank you for highlighting this omission in the manuscript. The data is presented as the ratio of 340/380 nm and the settings remained the same throughout all experimental groups. Background fluorescence was subtracted from all signals. This has now been added to the methods section.

Line 233: The settings remained the same throughout all experimental groups and background fluorescence was subtracted from all signals.

Cardiomyocytes were paced at 1 Hz, which is well below the normal heart rate of a rat in vivo. Please indicate the rationale for this.

1 Hz is a standard rate for most isolated cell studies regardless of species (Palmer et al., 1998; Curl et al., 2001; Kujala et al., 2012). Though this is indeed far less than the resting heart rate of a rat, it is not unusual to perform field stimulation studies at this pace, even for animals with a faster heart rate than rats, such as mice. One of the main reasons to support 1 Hz as a standard rate in experimental studies across species, is that faster stimulation using field stimulation can be technically problematic and result in failed stimulation events/missed beats. Calcium homeostasis in whole hearts stimulated at physiological rates will be investigated in future studies but were beyond the scope of the current project.

As calibration for determination of Ca²⁺ was not done (i.e. the y-axis in figure 4 is not [Ca²⁺], but fluorescence), please change the language in the results to indicate the actual measure used, and not the calcium concentration. For example, it would be more accurate to state "peak amplitude was decreased, indicating lower systolic calcium," rather than "Systolic Ca²⁺ was reduced." Additionally, the y-axis in figure 2 should be changed to F 340/380 or Fura 2 340/380.

We thank the Reviewer for the suggestion and have changed the language around fluorescence measures for clarity both in the text and figures using "F 340/380" throughout.

The figure legend should indicate the n/group, not the total n. n=10-12 animals per group is sufficient.

We have added n of animals within each group alongside the total number of cells for clarity in all figures and tables.

Representative myocytes in Fig.6 should be labeled in the figure image.

The myocytes in Figure 6 have now been labelled from their respective treatment groups.

Line 414: The thickened pulmonary artery does not necessarily promote PH, but is an indicator.

We have altered the language here to reflect thickening of the pulmonary artery as an indicator of hypertension rather than the cause of hypertension.

Line 479: For example, previous work has shown that offspring from hypoxic pregnancies have a thickened pulmonary artery, an indicator of pulmonary hypertension (Papamatheakis et al., 2013; Ding et al., 2020).

Please describe how, in this model, there may be secondary effects as the mother is also receiving reduced oxygen.

Thank you for this comment. We have expanded the Discussion to address this possibility.

Line 466: Since the mother in this rodent experimental model is exposed to hypoxia, it could be argued that some of the effects of hypoxic pregnancy programming an increased risk of cardiac dysfunction in the adult offspring may be due to maternal effects in addition to direct effects of chronic hypoxia on the developing fetal cardiovascular system. We have previously reported that this model of hypoxic pregnancy in rats is sufficient to lead to an increase in maternal haematocrit (Giussani et al., 2012) and the subtle induction of placental molecular indices of oxidative stress by the end of pregnancy (Richter et al., 2012). Indeed, maternal treatment with other antioxidants, such as vitamin C, in this model of maternal hypoxia protects against enhanced placental oxidative stress (Richter et al., 2012). Melatonin has potent antioxidant effects, which have been shown to protect against placental oxidative stress in other models of compromised pregnancy in rats (Richter et al., 2009), it is being trialled clinically for protecting against fetal brain damage and low birth weight in human pregnancy affected by fetal growth restriction (Palmer et al., 2019), and it is also known to promote an increase in umbilical blood flow by quenching excess free radical production and increasing the bioavailability of nitric oxide (Thakor et al., 2010). Therefore, some of the protective effects on programmed cardiac dysfunction in the adult offspring of maternal treatment with melatonin in hypoxic pregnancy in this model may be due to similar effects at the level of the placenta, protecting uteroplacental perfusion. Future integrative programmes of work, outside the scope of the present study, will address this possibility, combining studies of maternal cardiovascular function, uteroplacental vascular reactivity and placental molecular biology.

I am not sure I agree with the statement "between compensated and decompensated phases of heart failure." The degree of hypertrophy or other measures demonstrated here do not indicate remodeling close to actual models of heart failure, nor is it demonstrated that the rats would continue towards decompensation given the right stress (time or injury, for example). This is also true of the statement that MPI is "used clinically to diagnose cardiovascular disorders including dilated cardiomyopathy, ischemic heart disease and heart failure..." The authors should consider removing this statement as it implies that a statistical reduction in one measure is indicative of a disease state.

We thank the reviewer for this suggestion and have softened the tone in this section to reflect a predisposition of disease rather than heart failure itself. We have also removed the suggested sentence about myocardial performance index.

Line 488: Hence at 4 months old, equivalent to early adulthood in humans, the rats in our study appear to have a predisposition to developing compensated followed by decompensated phases of heart failure (Adusumalli & Mazurek, 2017).

The statement "Clinically, this finding suggests not all mothers should be treated, but antenatal melatonin therapy should only be applied once a diagnosis of hypoxic pregnancy has been made" should be removed, as this work does not demonstrate clinical outcomes nor consider the timing of administration.

We have rephrased this sentence to more accurately reflect the outcomes of the study, suggesting that; given our results, melatonin needs further investigation in the clinic as to, dosage and timing appropriateness for all pregnancies.

Line 488: Given the results of our study, clinical trials should consider further investigation into the dosage, timing and appropriateness of melatonin use in all mothers, and that antenatal melatonin therapy may only be appropriate for use once a diagnosis of hypoxic pregnancy has been made.

Referee #2:

This is an interesting study examining the impact of maternal hypoxia during most the pregnancy period (days 6 to 20) on adult offspring cardiac function. Both systolic and diastolic function were evaluated using echocardiography and isolated cardiomyocytes were used to assess Ca²⁺ transients.

The main findings corroborate previous studies by the authors that hypoxia during pregnancy induces long-lasting effects on the offspring cardiovascular. The authors also show in the present study that treatment of the mothers during pregnancy with melatonin prevented (or at least attenuated) most of the alterations observed in cardiac function in vivo and in isolated cardiomyocytes. Lastly, modest sex differences were observed when comparing male and female offspring.

Comments:

1) Lines 393 to 396: the authors state that the changes observed in Ca²⁺ handling are consistent with systolic and diastolic dysfunction assessed by echocardiography, including increased cardiac output. Why would cardiac output be increased by any of the alterations observed in "cardiac" Ca²⁺ in offspring from hypoxic pregnancies?

Cardiac output is not determined by the heart, unless the heart is in failure. Thus, the heart can, under certain circumstances limit cardiac output, but would never cause sustained increases in cardiac output.

Thank you for this comment. We agree. We have reviewed the Discussion carefully to remove any indication of a link between cardiac output and alterations in "cardiac" Ca²⁺ in offspring from hypoxic pregnancies.

2) The discussion about potential mechanisms responsible for the beneficial effect of melatonin during pregnancy to protect against the deleterious effects of hypoxia is very modest and basically concentrates on melatonin "antioxidant" actions. The Discussion section would benefit from a more broad and speculative discussion on potential mechanisms in addition to anti-oxidative properties on melatonin. Is there strong evidence that offspring from hypoxic pregnancies show increased ROS generation or reduced antioxidant capacity? These parameters, perhaps, could have been measured in isolated cardiomyocytes (although,

this reviewer is not suggesting that this should be included in the present study).

We thank the reviewer for the recommendation. We have expanded the discussion, as requested.

Line 466: Since the mother in this rodent experimental model is exposed to hypoxia, it could be argued that some of the effects of hypoxic pregnancy programming an increased risk of cardiac dysfunction in the adult offspring may be due to maternal effects in addition to direct effects of chronic hypoxia on the developing fetal cardiovascular system. We have previously reported that this model of hypoxic pregnancy in rats is sufficient to lead to an increase in maternal haematocrit (Giussani et al., 2012) and the subtle induction of placental molecular indices of oxidative stress by the end of pregnancy (Richter et al., 2012). Indeed, maternal treatment with other antioxidants, such as vitamin C, in this model of maternal hypoxia protects against enhanced placental oxidative stress (Richter et al., 2012). Melatonin has potent antioxidant effects, which have been shown to protect against placental oxidative stress in other models of compromised pregnancy in rats (Richter et al., 2009), it is being trialled clinically for protecting against fetal brain damage and low birth weight in human pregnancy affected by fetal growth restriction (Palmer et al., 2019), and it is also known to promote an increase in umbilical blood flow by quenching excess free radical production and increasing the bioavailability of nitric oxide (Thakor et al., 2010). Therefore, some of the protective effects on programmed cardiac dysfunction in the adult offspring of maternal treatment with melatonin in hypoxic pregnancy in this model may be due to similar effects at the level of the placenta, protecting uteroplacental perfusion. Future integrative programmes of work, outside the scope of the present study, will address this possibility, combining studies of maternal cardiovascular function, uteroplacental vascular reactivity and placental molecular biology.

3) Was blood pressure measured in the offspring? The major cause of systolic and diastolic dysfunction in humans is elevated blood pressure.

Thank you for this comment. We have previously reported that this model of hypoxic pregnancy in rats does not programme alterations in basal arterial blood pressure in 4-month adult offspring, as in this study (Lakshman et al., 2021; Spiroski et al., 2021). In those studies arterial blood pressure was measured for several days in chronically instrumented adult rat offspring with indwelling vascular catheters. Arterial blood pressure was not measured in this study. To address the Reviewer's comment and highlight this point, we have expanded the Discussion.

Line 427: We have previously reported that this model of hypoxic pregnancy in rats does not programme alterations in basal arterial blood pressure measured via chronically instrumented indwelling catheters in 4-month adult offspring (Lakshman et al., 2021; Spiroski et al., 2021). Therefore, effects of adult offspring ventricular remodelling programmed by developmental hypoxia are independent of effects on the adult offspring cardiac afterload. The ventricular remodelling is therefore more likely to represent developmentally programmed effects triggered by the fetal brain sparing response to hypoxic pregnancy (Giussani, 2016).

- Curl CL, Wendt IR & Kotsanas G. (2001). Effects of gender on intracellular [Ca²⁺] in rat cardiac myocytes. *Pflugers Arch* **441**, 709-716.
- Kujala K, Ahola A, Pekkanen-Mattila M, Ikonen L, Kerkelä E, Hyttinen J & Aalto-Setälä K. (2012). Electrical Field Stimulation with a Novel Platform: Effect on Cardiomyocyte Gene Expression but not on Orientation. *Int J Biomed Sci* **8**, 109-120.
- Lakshman R, Spiroski AM, McIver LB, Murphy MP & Giussani DA. (2021). Noninvasive Biomarkers for Cardiovascular Dysfunction Programmed in Male Offspring of Adverse Pregnancy. *Hypertension* **78**, 1818-1828.
- Palmer BM, Thayer AM, Snyder SM & Moore RL. (1998). Shortening and [Ca²⁺] dynamics of left ventricular myocytes isolated from exercise-trained rats. *Journal of Applied Physiology* **85**, 2159-2168.
- Spiroski AM, Niu Y, Nicholas LM, Austin-Williams S, Camm EJ, Sutherland MR, Ashmore TJ, Skeffington KL, Logan A, Ozanne SE, Murphy MP & Giussani DA. (2021). Mitochondria antioxidant protection against cardiovascular dysfunction programmed by early-onset gestational hypoxia. *FASEB J* **35**, e21446.

Dear Dr Galli,

Re: JP-RP-2024-287325R1 "Maladaptive Cardiomyocyte Calcium Handling in Adult Offspring of Hypoxic Pregnancy: Protection by Antenatal Maternal Melatonin" by Mitchell C Lock, Olga Patey, Kerri LM Smith, Youguo Niu, Ben Jaggs, Andrew W Trafford, Dino A Giussani, and Gina Lucia Jane Galli

We are pleased to tell you that your paper has been accepted for publication in The Journal of Physiology.

Authors should note that it is too late at this point to offer corrections prior to proofing. Major corrections at proof stage, such as changes to figures, will be referred to the Editors for approval before they can be incorporated. Only minor changes, such as to style and consistency, should be made at proof stage. Changes that need to be made after proof stage will usually require a formal correction notice.

If you would like to receive our 'Research Roundup', a monthly newsletter highlighting the cutting-edge research published in The Physiological Society's family of journals (The Journal of Physiology, Experimental Physiology and Physiological Reports), please click this link, fill in your name and email address and select 'Research Roundup': <https://www.physoc.org/journals-and-media/membernews/>.

Yours sincerely,

Laura Bennet
Senior Editor
The Journal of Physiology

P.S. - You can help your research get the attention it deserves! Check out Wiley's free Promotion Guide for best-practice recommendations for promoting your work at www.wileyauthors.com/eeo/guide. You can learn more about Wiley Editing Services which offers professional video, design, and writing services to create shareable video abstracts, infographics, conference posters, lay summaries, and research news stories for your research at www.wileyauthors.com/eeo/promotion.

IMPORTANT NOTICE ABOUT OPEN ACCESS: To assist authors whose funding agencies mandate public access to published research findings sooner than 12 months after publication, The Journal of Physiology allows authors to pay an Open Access (OA) fee to have their papers made freely available immediately on publication.

You can check if your funder or institution has a Wiley Open Access Account here: <https://authorservices.wiley.com/author-resources/Journal-Authors/licensing-and-open-access/open-access/author-compliance-tool.html>.

EDITOR COMMENTS

Reviewing Editor:

Thank you for your alterations, there are no further comments to address. Congratulations on your newest paper.

REFEREE COMMENTS

Referee #1:

Thank you for the comments, the manuscript has been revised to appropriately address previous concerns.

Referee #2:

No additional comments.

1st Confidential Review

17-Sep-2024